# The Delta Rule Dominates: A Factorial Analysis of Decay in Linear Attention

## Abstract

In a controlled fixed-budget study of linear attention decay mechanisms, we find that the delta rule is the largest observed factor under our protocol. We evaluate seven linear-attention variants plus a standard softmax baseline on two datasets (TinyStories, WikiText-103) with 3 random seeds each at 18M parameters, and on TinyStories with 3 seeds at 125M and 5 variants at 42M. The data-independent subset, where decay gates are learned parameters rather than token-conditioned functions, forms a complete $2 \times 2$ comparison over granularity (scalar vs. channel-wise) and delta rule (absent vs. present), while the data-dependent variants provide additional controlled pairwise comparisons. We measure the delta rule's dominance by rank separation and loss gaps: all delta-rule variants (ranks 1–4) beat every model without a delta-rule write, including the softmax baseline (ranks 5–8), at 18M (both datasets) and 125M (TinyStories), and at 125M the worst delta variant beats the best model without a delta-rule write by 0.057 nats, $6\times$ larger than the top-3 delta spread. A *granularity×delta interaction* provides the second key finding within the studied LM settings: channel-wise decay hurts without the delta rule but helps with it. Within the top-4 delta variants, rankings are scale-dependent: data-independent *StaticChannelDelta* leads at 18M, while data-dependent KDA overtakes it at 125M as the top-3 gap compresses to 0.009 nats. A single-seed synthetic recall probe suggests a *task-dependent* tradeoff: scalar+delta variants achieve above 90% mean exact retrieval accuracy under the 15K-step budget, while channel-wise+delta variants plateau near 27%, indicating that channel-wise decay may trade retrieval precision for representational richness.

## 1 Introduction

Linear attention (Katharopoulos et al., 2020) replaces the softmax attention mechanism with a recurrent state matrix $\mathbf{S} \in \mathbb{R}^{d \times d}$, which serves as an associative memory. Let $\mathbf{q}_t$, $\mathbf{k}_t$, and $\mathbf{v}_t$ denote the projected query, key, and value vectors at position $t$ (the full projection equations appear in Section 3). We write the recurrent update shared by the linear-attention variants as

$$\mathbf{S}_t = \mathbf{G}_t \odot \mathbf{S}_{t-1} + \boldsymbol{\kappa}_t \otimes \mathbf{u}_t, \tag{1}$$

$$\mathbf{o}_t = \mathbf{S}_t^\top \phi(\mathbf{q}_t), \tag{2}$$

with initial state $\mathbf{S}_0 = \mathbf{0}$ for each sequence and head. Here $\mathbf{G}_t$ is a scalar or channel-wise decay gate, $\boldsymbol{\kappa}_t$ is the key written to the recurrent state, $\mathbf{u}_t$ is the value-like write vector, $\otimes$ denotes the outer product, and $\phi$ is the query feature map. This formulation enables $O(1)$ memory and $O(n)$ compute during inference, compared to $O(n^2)$ for softmax attention.

The *delta rule* is the corrective write mechanism studied in this paper. Ordinary non-delta variants write $\boldsymbol{\kappa}_t = \phi(\mathbf{k}_t)$ and $\mathbf{u}_t = \mathbf{v}_t$. Delta-rule variants instead write with the normalized key $\boldsymbol{\kappa}_t = \hat{\mathbf{k}}_t$ and the prediction-error vector

$$\mathbf{u}_t = \mathbf{v}_t - \mathbf{S}_{t-1}^\top \hat{\mathbf{k}}_t, \qquad \hat{\mathbf{k}}_t = \mathbf{k}_t / \|\mathbf{k}_t\|_2. \tag{3}$$

Thus, the model writes residual information not already predicted by the current memory rather than blindly adding a new outer product.

The critical design choice is **how the decay works**: it determines what information is retained and what is forgotten. Two recent lines of work have independently identified improvements along orthogonal design axes that we call *granularity* (scalar vs. channel-wise gating) and *conditioning* (data-dependent vs. data-independent gates):

**Axis 1: Granularity.** Many linear attention models—including Gated DeltaNet (GDN; Yang et al. 2025) and Mamba2 (Dao & Gu, 2024)—use a single scalar decay rate per head, applied uniformly across all dimensions of the state matrix. (The published GLA (Yang et al., 2023) uses a richer low-rank per-element gate; our controlled study uses its scalar special case to isolate the granularity axis.) KDA (Kimi Delta Attention; Zhang et al. 2025) instead uses a *per-channel* gate vector $\mathbf{G}_t \in \mathbb{R}^d$, allowing each dimension to decay at its own rate. This enables multi-timescale memory: some channels retain long-range facts (high gate values near 1) while others track short-range context (low gate values). On long-context benchmarks, Kimi Linear (built on KDA) substantially outperforms its GDN-based counterpart on MRCR (29.6 vs. 23.9 at 128k context; Zhang et al. 2025, Table 5), suggesting a clear benefit from channel-wise structure. (However, this advantage is task-dependent: in Section 4.8, we show that on exact key-value binding retrieval at shorter ranges, scalar+delta variants *outperform* channel-wise+delta variants, suggesting that channel-wise gating aids long-range persistence but can hinder precise recall.)

**Axis 2: Conditioning.** HALO (Chen et al., 2026) observed that in their hybrid architecture (HypeNet), Lightning Attention, which uses a single fixed scalar decay per head ($\gamma_h = \exp(-2^{-8h/H})$), achieves better length generalization than data-dependent alternatives (GLA Yang et al., 2023, Gated DeltaNet Yang et al., 2025, Mamba2 Dao & Gu, 2024). The authors suggest that data-dependent forget gates "may result in poor length generalization," as argued by Chen et al. (2025). Importantly, HALO's finding is in the context of hybrid models where $\sim$25% of layers are still softmax attention, and the authors attribute their length generalization to the *combination* of Lightning Attention and HyPE (their positional encoding scheme), not to data-independent gating alone. Additionally, HALO's Lightning Attention uses a simple outer-product write (no delta rule), so its results do not directly speak to the interaction of static gating with the delta rule.

The design space of decay mechanisms can be organized as a $2 \times 2$ matrix. Prior work tested three of four quadrants; the data-independent column lacked systematic evaluation, especially in conjunction with the delta rule:

Table 1: Prior decay-design quadrants. Here "data" means the input token sequence as represented by the layer activations $\mathbf{x}_t$. "Data-dependent" means the decay gate is computed from the current token representation $\mathbf{x}_t$; "data-independent" means the gate is fixed or learned as a model parameter independent of the input sequence. The recurrent state $\mathbf{S}_{t-1}$ in Eq. 1 is shared memory used by all variants and is not what this conditioning axis refers to.

| | **Data-Dependent** | **Data-Independent** |
|---|---|---|
| **Scalar gate** | GDN, DeltaNet, Mamba2 | Lightning Attention (HALO); scalar only, no delta rule |
| **Channel-wise gate** | GLA[†], KDA | *(untested)* |

[†]The published GLA uses a low-rank per-element gate, placing it in the channel-wise row. Our factorial study uses GLA's scalar special case to isolate the granularity axis (see Table 2).

We fill **both** data-independent quadrants and cross them with the delta rule, yielding four new conditions whose names encode their design choices: "Scalar" vs. "Channel" (granularity), "Static" (data-independent conditioning), and "Delta" (delta rule). Combined with controlled data-dependent variants and standard softmax attention, this produces eight total evaluated model variants: seven linear-attention variants plus the softmax baseline (Table 2). The naming convention is systematic: *Standard* is the softmax baseline; our *GLA* variant uses the scalar special case of GLA's gate (Yang et al., 2023), and *DeltaNet* adds the delta rule to the same scalar gate; *KDA* extends this to data-dependent channel-wise gating; and the four "Static" variants are our new data-independent conditions.

Table 2: Seven linear-attention variants plus a softmax baseline. Bold rows are new data-independent conditions introduced in this work; others are controlled reimplementations of existing designs.

| Variant | Granularity | Conditioning | Delta rule? |
|---|---|---|---|
| Standard | – (softmax) | – | – |
| GLA | Scalar | Data-dependent | No |
| DeltaNet | Scalar | Data-dependent | Yes |
| KDA | Channel-wise | Data-dependent | Yes |
| **ScalarStatic** | **Scalar** | **Data-independent** | **No** |
| **ScalarStaticDelta** | **Scalar** | **Data-independent** | **Yes** |
| **StaticChannel** | **Channel-wise** | **Data-independent** | **No** |
| **StaticChannelDelta** | **Channel-wise** | **Data-independent** | **Yes** |

One cell of the $2 \times 2 \times 2$ cube is absent: channel-wise, data-dependent, no delta (occupied by RWKV; Peng et al. 2024), which we do not reimplement. The full cube is therefore not a complete $2 \times 2 \times 2$ factorial design. Our factorial decomposition (Tables 7–8) uses the four data-independent variants, where all four granularity×delta cells are present; the data-dependent variants provide controlled pairwise comparisons. Figure 1 visualizes the design space.

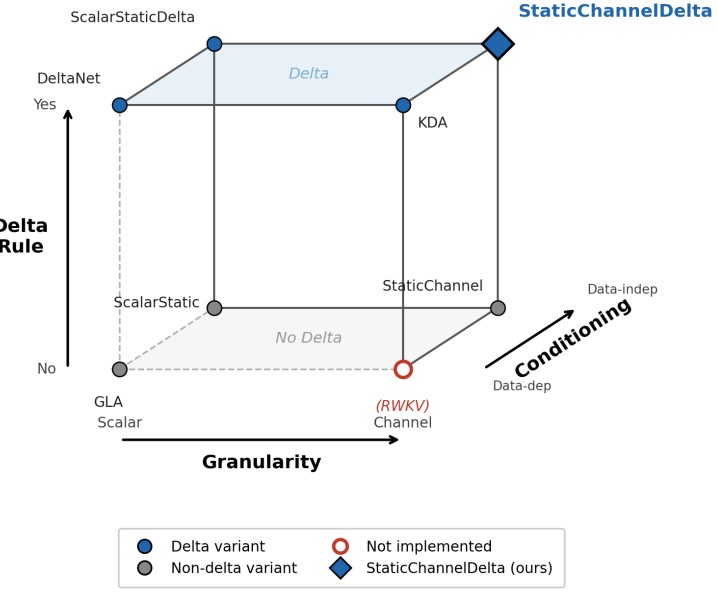

Figure 1: The $2 \times 2 \times 2$ design space of decay mechanisms. Each vertex represents a combination of granularity (scalar vs. channel-wise), conditioning (data-dependent vs. data-independent), and delta rule (present vs. absent). Filled blue circles are delta-rule variants; filled gray circles are non-delta variants; the hollow red circle marks the unimplemented cell (channel-wise, data-dependent, no delta), occupied by RWKV in the literature. Standard softmax attention lies outside this cube.

The concurrent work of Qin et al. (2025) systematically evaluates three quadrants of this space but leaves the channel-wise + data-independent combination untested and does not incorporate the **delta rule** (Schlag et al., 2021), which replaces the ordinary outer-product write with the prediction-error write in Eq. 3.

Our contribution is **identifying interaction structure under a controlled fixed-budget protocol**, not proposing a single best architecture or establishing convergence-optimal rankings. We evaluate seven

linear-attention variants plus a softmax baseline under a shared backbone, optimizer, token budget, and random-seed protocol. This reveals three findings. First, the **delta rule is the strongest observed factor in our controlled comparisons**: measured by rank separation and loss gaps, all delta-rule variants outperform every model without a delta-rule write, including the softmax baseline, at 18M on both datasets and at 125M on TinyStories, where the worst delta variant beats the best model without a delta-rule write by 0.057 nats. Second, in the complete data-independent $2 \times 2$ subspace, we observe a **granularity×delta interaction**: within the studied LM settings, channel-wise granularity hurts without the delta rule but helps with it, consistent with the delta rule being especially useful when multiple decay timescales are present. Third, a single-seed synthetic recall benchmark suggests a **task-dependent tradeoff**: scalar+delta variants achieve above 90% mean exact associative recall accuracy under the 15K-step probe, while channel-wise+delta variants plateau near 27%, consistent with a tradeoff between retrieval precision and the richer multi-timescale representations that benefit language modeling. Within the delta variants, the best configuration is scale-dependent: data-independent StaticChannelDelta leads at 18M, while data-dependent KDA overtakes it at 125M. We do not claim superiority over fully-optimized published systems; our controlled design reduces implementation differences to isolate the effect of each design axis within this protocol.

## 2 Related Work

### 2.1 Linear Attention and State-Space Models

Linear attention (Katharopoulos et al., 2020) reformulates attention as a recurrence over an outer-product state matrix. Subsequent work introduced gating to control the accumulation: GLA (Yang et al., 2023) adds a data-dependent gate parameterized as a low-rank per-element diagonal matrix, while RetNet (Sun et al., 2023) uses fixed exponential decay.

State-space models (SSMs) like Mamba (Gu & Dao, 2023) and Mamba2 (Dao & Gu, 2024) can be viewed through the same lens: they maintain a recurrent state with input-dependent gating. Mamba2 in particular unifies SSMs with linear attention by showing both are instances of the same "structured state space duality."

### 2.2 Delta Rule in Linear Attention

The delta rule (Schlag et al., 2021), defined in Eq. 3, replaces an ordinary value write with a corrective prediction-error write. It subtracts the value currently predicted by memory before writing, analogous to Widrow–Hoff learning in associative memory. DeltaNet and subsequent work (Yang et al., 2025) show that this dramatically improves recall in linear attention. Key normalization is required for stability, as predictions formed from unnormalized keys can grow unboundedly.

### 2.3 Channel-Wise Gating

KDA (Zhang et al., 2025) introduces *data-dependent* per-dimension gating via a diagonal matrix: $\mathbf{S}_t = \mathrm{diag}(\boldsymbol{\alpha}_t) \cdot \mathbf{S}_{t-1} + \hat{\mathbf{k}}_t \otimes \mathrm{error}_t$, where $\boldsymbol{\alpha}_t \in \mathbb{R}^d$ is computed from the current input token via a learned projection and $\mathrm{error}_t$ is the delta-rule prediction error. This allows different state dimensions to decay at different rates, enabling multi-timescale memory within a single head.[1]

---

[1]Our implementations of KDA and DeltaNet simplify the full published architectures in two ways that are uniform across all variants: (1) the delta-rule error is computed from the un-decayed state $(\mathbf{v}_t - \mathbf{S}_{t-1}^\top \hat{\mathbf{k}}_t)$ rather than the decayed state $(\mathbf{v}_t - (\mathrm{diag}(\boldsymbol{\alpha}_t)\mathbf{S}_{t-1})^\top \hat{\mathbf{k}}_t)$, and (2) the separate scalar learning rate $\beta_t$ from the original formulations is omitted (effectively $\beta_t = 1$). Additionally, we use a single full-rank linear projection ($\mathbf{x}_t W_g \in \mathbb{R}^d$) for the channel-wise gate rather than the low-rank factorization ($\mathbf{x}_t W_1 W_2$, with $W_1 \in \mathbb{R}^{D \times r}$, $W_2 \in \mathbb{R}^{r \times d}$, $r \ll D$) used in the Kimi Linear architecture to reduce parameter count. Since these simplifications are applied consistently across all delta-rule variants (DeltaNet, KDA, ScalarStaticDelta, StaticChannelDelta), they reduce implementation confounds and support controlled comparisons, though they do not rule out the possibility that some simplifications interact differently with specific variants.

### 2.4 Data-Independent Decay

Lightning Attention (used in HALO; Chen et al. 2026) assigns each head a fixed scalar decay $\gamma_h = \exp(-2^{-8h/H})$, producing an exponential spectrum of *timescales* (i.e., effective memory half-lives) across heads: head 0 forgets quickly while head $H-1$ retains information over hundreds of tokens. It uses an ordinary outer-product write (no delta rule): $\mathbf{S}_t = \gamma_h \mathbf{S}_{t-1} + \phi(\mathbf{k}_t) \otimes \mathbf{v}_t$ in the notation of Eq. 1. In HALO's hybrid architecture (HypeNet), Lightning Attention achieves better length generalization than data-dependent RNN mixers (GLA, GDN, Mamba2, RWKV7), though the authors attribute this to the *combination* of Lightning Attention and their HyPE positional encoding scheme. HALO also reports that KDA failed to converge during distillation (Appendix G.2), suggesting practical stability issues with data-dependent channel-wise gating at scale.

### 2.5 Design Space of Decay

Qin et al. (2025) systematically study decay parameterization along three axes: conditioning strategy (static/fixed, trainable but data-independent, or input-conditional/data-dependent), granularity (scalar, vector), and initialization. They test three of the four quadrants in the {scalar, vector} × {data-dependent, data-independent} grid but leave vector + data-independent untested. They also do not consider the delta rule.

### 2.6 Positioning of Other Architectures

Several prominent architectures map directly onto our factorial grid. RetNet (Sun et al., 2023) uses a fixed scalar decay per head, occupying the same quadrant as our ScalarStatic variant. Mamba2 (Dao & Gu, 2024) uses data-dependent scalar gating (the "structured state space duality" with $A_t$ diagonal), placing it in the GLA quadrant. RWKV (Peng et al., 2024) similarly uses data-dependent channel-wise gating without the delta rule, corresponding to the one cube cell we leave unimplemented. We do not compare against fully optimized versions of these architectures; our goal is to isolate the effect of each design axis (granularity, conditioning, delta rule) within a shared backbone rather than to confound those axes with engineering differences (kernel implementations, learning rate schedules, model-specific tuning).

In summary, the design space has three binary axes (granularity, conditioning, and the delta rule), but only a subset of combinations has been tested. We next describe the controlled variants used in our comparisons.

## 3 Method

### 3.1 Background: Unified Recurrence

All variants we study share the same backbone. Given input $\mathbf{x} \in \mathbb{R}^{B \times T \times D}$, we project to queries, keys, and values:

$$\mathbf{q} = \mathbf{x}W_q, \quad \mathbf{k} = \mathbf{x}W_k, \quad \mathbf{v} = \mathbf{x}W_v \tag{4}$$

split into $H$ heads of dimension $d = D/H$, and maintain a state $\mathbf{S} \in \mathbb{R}^{d \times d}$ per head using Eqs. 1–2. Throughout the paper, $\mathbf{k}_t$ denotes the projected key, $\hat{\mathbf{k}}_t$ denotes its $\ell_2$-normalized form used by delta-rule variants, and $\boldsymbol{\kappa}_t$ denotes the key actually written to the recurrent state. The feature map $\phi$ is a nonlinear transformation that maps queries into a non-negative space so that the inner product $\mathbf{S}_t^\top \phi(\mathbf{q}_t)$ acts as an unnormalized kernel attention. We use $\phi(\mathbf{x}) = \text{ELU}(\mathbf{x}) + 1$ (Katharopoulos et al., 2020), which maps each element $x$ to $\max(0, x) + \min(0, e^x - 1) + 1$, ensuring all outputs are positive. This is applied to queries in all variants. For non-delta variants, the same ELU+1 feature map is applied to projected keys, giving $\boldsymbol{\kappa}_t = \phi(\mathbf{k}_t)$. For delta-rule variants, the write key is the normalized key $\boldsymbol{\kappa}_t = \hat{\mathbf{k}}_t$, as in Eq. 3. When $\mathbf{G}_t$ is a vector ($\in \mathbb{R}^d$), the product $\mathbf{G}_t \odot \mathbf{S}_{t-1}$ denotes row-wise scaling, i.e., $\text{diag}(\mathbf{G}_t) \mathbf{S}_{t-1}$; when $\mathbf{G}_t$ is scalar, it broadcasts across all entries.

### 3.2 StaticChannelDelta

StaticChannelDelta combines three ingredients:

**1. Channel-wise decay (from KDA).** The gate $\mathbf{\Gamma} \in \mathbb{R}^{H \times d}$ has one value per dimension, allowing each row of $\mathbf{S}$ to decay at its own rate. (We use $\mathbf{\Gamma}$ rather than $\mathbf{G}_t$ to emphasize that this gate does not depend on the input: unlike data-dependent gates, which are recomputed at every token, $\mathbf{\Gamma}$ is a fixed model parameter applied identically at every position.)

**2. Data-independent conditioning (from Lightning Attention).** The gate $\mathbf{\Gamma}$ is a *learnable parameter* that does not depend on the input token. It is stored as unconstrained logits $\boldsymbol{\ell} \in \mathbb{R}^{H \times d}$ and passed through the sigmoid function $\sigma(x) = 1/(1 + e^{-x})$ to produce decay rates in $(0, 1)$: $\mathbf{\Gamma} = \sigma(\boldsymbol{\ell})$. Because $\boldsymbol{\ell}$ is a model parameter (like a weight matrix), $\mathbf{\Gamma}$ is fixed for a given trained model: it does not change with the input sequence, so the decay behavior is identical whether the model processes 512 or 100,000 tokens.

**3. Delta rule (from DeltaNet/KDA).** For this variant, the write key in Eq. 1 is the normalized key $\boldsymbol{\kappa}_t = \hat{\mathbf{k}}_t$, and the write vector is the prediction-error vector from Eq. 3.

Substituting these choices and the static channel-wise gate $\mathbf{\Gamma}$ into Eq. 1 gives:

$$\mathbf{S}_t = \mathbf{\Gamma} \odot \mathbf{S}_{t-1} + \hat{\mathbf{k}}_t \otimes (\mathbf{v}_t - \mathbf{S}_{t-1}^\top \hat{\mathbf{k}}_t) \tag{5}$$

with output $\mathbf{o}_t = \mathbf{S}_t^\top \phi(\mathbf{q}_t)$, where $\phi$ is the ELU+1 feature map applied to queries.

## 3.3 Initialization: Exponential Timescale Spectrum

Following Lightning Attention, we initialize each channel's decay to form an exponential spectrum. Denoting the $i$-th entry of the per-head gate vector $\mathbf{\Gamma}$ as $\gamma_i$ (so that $\mathbf{\Gamma} = [\gamma_0, \gamma_1, \ldots, \gamma_{d-1}]$ for each head):

$$\gamma_i = \exp\left(-2^{-8i/d}\right), \quad i = 0, 1, \ldots, d-1 \tag{6}$$

This produces initial decay rates ranging from $\gamma_0 \approx 0.37$ (fast forgetting) to $\gamma_{d-1} \approx 0.996$ (slow forgetting). Channel 0 has a half-life of $\sim$1 token while channel $d-1$ (with $d=64$) has a half-life of $\sim$163 tokens.

Since we parameterize gates as $\mathbf{\Gamma} = \sigma(\boldsymbol{\ell})$, we need to convert the desired initial decay rates $\gamma_i$ to logit space. Applying the inverse sigmoid (the logit function $\sigma^{-1}(p) = \log(p/(1-p))$):

$$\ell_i = \log\left(\frac{\gamma_i}{1 - \gamma_i}\right) \tag{7}$$

This initialization is shared across heads but each head's $\boldsymbol{\ell}$ is an independent parameter that can diverge during training.

## 3.4 ScalarStatic and ScalarStaticDelta

To enable a complete data-independent $2 \times 2$ factorial decomposition, we also implement two scalar-gate, data-independent variants that mirror Lightning Attention / RetNet:

**ScalarStatic** uses a single fixed scalar decay per head: $\mathbf{S}_t = \gamma_h \mathbf{S}_{t-1} + \phi(\mathbf{k}_t) \otimes \mathbf{v}_t$, where $\gamma_h = \sigma(\ell_h)$ is a learnable parameter initialized from the exponential spectrum $\gamma_h = \exp(-2^{-8h/H})$, adapting the index range to heads rather than channels ($h = 0, \ldots, H-1$ instead of $i = 0, \ldots, d-1$). Each head has its own independent decay rate, so a 4-head model has 4 learnable scalars. This is equivalent to Lightning Attention with learnable (rather than fixed) decay rates.

**ScalarStaticDelta** adds the delta rule: $\mathbf{S}_t = \gamma_h \mathbf{S}_{t-1} + \hat{\mathbf{k}}_t \otimes (\mathbf{v}_t - \mathbf{S}_{t-1}^\top \hat{\mathbf{k}}_t)$. This tests whether the delta rule's benefit is specific to channel-wise granularity or extends to scalar gating as well.

## 3.5 Comparison of All Variants

Table 3 summarizes the seven linear-attention variants plus the softmax baseline. The "Key transform" column indicates how keys are preprocessed before being used in the state update: non-delta variants use the ELU+1 feature map (matching queries) to produce non-negative kernel features, while delta-rule variants use

$\ell_2$-normalization (required for stability of the corrective error, as discussed in Section 3.1). The "Quadrant" column encodes the granularity×conditioning combination: DD = data-dependent, DI = data-independent. Note that GLA and DeltaNet share a quadrant (data-dependent scalar) but differ in whether the delta rule is used, enabling direct isolation of the delta rule's effect within that quadrant.

Table 3: Summary of seven linear-attention variants plus the softmax baseline. Bold variant names are the new data-independent conditions introduced in this work.

| Variant | Gate shape | Data-dep? | Delta rule? | Key transform | Quadrant |
|---|---|---|---|---|---|
| Standard | – (softmax) | – | – | – | Baseline |
| GLA | Scalar | Yes | No | ELU+1 | DD-Scalar |
| DeltaNet | Scalar | Yes | Yes | L2-norm | DD-Scalar |
| KDA | Channel-wise | Yes | Yes | L2-norm | DD-Channel |
| **ScalarStatic** | Scalar | No | No | ELU+1 | DI-Scalar |
| **ScalarStaticDelta** | Scalar | No | Yes | L2-norm | DI-Scalar |
| **StaticChannel** | Channel-wise | No | No | ELU+1 | DI-Channel |
| **StaticChannelDelta** | Channel-wise | No | Yes | L2-norm | DI-Channel |

## 4 Experiments

### 4.1 Setup

**Model.** Our primary experiments use a GPT-style transformer with 6 layers, 4 heads, $d_{\mathrm{model}} = 256$, totaling ~18M parameters. We additionally scale up to ~42M ($d_{\mathrm{model}} = 512$, 8 layers, 8 heads) and ~125M ($d_{\mathrm{model}} = 768$, 12 layers, 12 heads) in Section 4.10. The attention mechanism is the only component that varies between experiments; all other hyperparameters (FFN size, normalization, embedding) are identical across variants at each scale.

**Data.** To validate that our findings reflect properties of the *mechanism* rather than artifacts of a particular dataset, we train on multiple corpora of different character:

- **TinyStories** (Eldan & Li, 2023): Synthetic children's stories with simple grammar, limited vocabulary, and short narratives.

- **WikiText-103** (Merity et al., 2016): Real encyclopedic text from Wikipedia with complex syntax, diverse topics, and factual content.

All datasets use GPT-2 BPE tokenization (50257 vocab), 10M training tokens, ~200K–500K validation tokens, and sequence length 512. Using identical token budgets and hyperparameters across datasets isolates the effect of the decay mechanism from data-specific confounds.

**Training.** AdamW optimizer ($\beta_1$=0.9, $\beta_2$=0.95), learning rate $3 \times 10^{-4}$ with linear warmup (500 steps) and cosine decay. Weight decay 0.1, gradient clipping at 1.0, 3 epochs with batch size 8. We run each configuration with 3 random seeds (42, 123, 7) and report mean $\pm$ standard deviation.

**Evaluation metric.** All reported "validation loss" values are token-level cross-entropy (in nats), averaged over all tokens in the validation set under teacher forcing at the training sequence length (512). No padding is used; sequences are packed contiguously and split into fixed-length chunks.

**Hardware.** Apple M4 Max, 96GB RAM, MPS backend. All implementations are naive recurrent-loop PyTorch (no Triton kernels), prioritizing correctness and reproducibility over speed.

**Baseline faithfulness.** Our implementations of KDA and DeltaNet simplify the full published architectures in ways that are uniform across all variants (see footnote in Section 2.3): the delta-rule error uses the un-decayed state, the scalar learning rate $\beta_t$ is fixed at 1, and channel-wise gates use a full-rank projection. We therefore distinguish two levels of claims: (A) *controlled-comparison claims*: comparisons among our

variants within a consistent framework, where simplifications applied consistently to all applicable variants (i.e., all delta-rule variants share the same simplified error computation and $\beta_t{=}1$) reduce implementation confounds, though they do not rule out variant-specific interactions; and (B) *claims relative to prior work*: we do **not** claim to beat the fully-optimized published implementations of KDA or DeltaNet, which use chunkwise kernels, low-rank gate parameterizations, and tuned $\beta_t$ schedules. Our contribution is identifying the interaction structure of the design space, not establishing absolute state-of-the-art perplexity.

**Evaluation strategy.** Our goal is to identify which design axes matter and how they interact under a matched training budget, not to establish a new state-of-the-art system or convergence-optimal ranking. We therefore prioritize *controlled comparisons* (identical backbone, identical hyperparameters, multiple seeds and datasets) over raw scale. The complete data-independent $2 \times 2$ subset enables decomposition of granularity and delta-rule effects; the data-dependent variants provide additional controlled pairwise comparisons. We validate robustness via 3 seeds $\times$ 2 datasets at 18M, scale-up to 42M (single seed), and scale-up to 125M with 3 seeds to test whether the structural findings persist. We report all individual seed results to enable the reader to assess effect sizes relative to seed variance.

## 4.2 Results: TinyStories

Table 4 reports validation loss on TinyStories across 3 seeds. All four delta-rule variants (top 4) beat every model without a delta-rule write, including the softmax baseline (bottom 4), with StaticChannelDelta achieving the lowest mean loss.

Table 4: Validation loss on TinyStories (mean $\pm$ std across 3 seeds). Lower is better; bold denotes the best mean. The four delta-rule variants occupy the top four ranks.

| Model | Val Loss | Seed 42 | Seed 123 | Seed 7 | Gate Type | Delta? |
|---|---|---|---|---|---|---|
| **StaticChannelDelta** | **2.2591 $\pm$ 0.012** | 2.2429 | 2.2625 | 2.2718 | Ch, static | Yes |
| KDA | 2.2820 $\pm$ 0.020 | 2.3079 | 2.2772 | 2.2607 | Ch, learned | Yes |
| ScalarStaticDelta | 2.3015 $\pm$ 0.005 | 2.2939 | 2.3053 | 2.3052 | Sc, static | Yes |
| DeltaNet | 2.3222 $\pm$ 0.017 | 2.3465 | 2.3125 | 2.3076 | Sc, learned | Yes |
| Standard Attention | 2.3833 $\pm$ 0.001 | 2.3849 | 2.3825 | 2.3824 | – (softmax) | – |
| GLA | 2.4207 $\pm$ 0.004 | 2.4228 | 2.4152 | 2.4242 | Sc, learned | No |
| ScalarStatic | 2.4307 $\pm$ 0.005 | 2.4241 | 2.4325 | 2.4355 | Sc, static | No |
| StaticChannel | 2.5491 $\pm$ 0.014 | 2.5423 | 2.5365 | 2.5685 | Ch, static | No |

**Takeaway.** On TinyStories, the four delta-rule variants occupy the top four mean ranks, and the best model without a delta-rule write is separated from the worst delta-rule variant by 0.061 nats.

## 4.3 Results: WikiText-103

Table 5 reports the same experiment on WikiText-103, a more challenging corpus of encyclopedic text. The top-4 ranking is identical to TinyStories, confirming that the pattern is not dataset-specific.

Table 5: Validation loss on WikiText-103 (mean ± std across 3 seeds). Lower is better; bold denotes the best mean. The four delta-rule variants again occupy the top four ranks.

| Model | Val Loss | Seed 42 | Seed 123 | Seed 7 | Gate Type | Delta? |
|---|---|---|---|---|---|---|
| **StaticChannelDelta** | **4.9207 ± 0.001** | 4.9221 | 4.9195 | 4.9206 | Ch, static | Yes |
| KDA | 4.9353 ± 0.010 | 4.9248 | 4.9482 | 4.9328 | Ch, learned | Yes |
| ScalarStaticDelta | 4.9704 ± 0.009 | 4.9623 | 4.9821 | 4.9667 | Sc, static | Yes |
| DeltaNet | 4.9869 ± 0.008 | 4.9946 | 4.9753 | 4.9907 | Sc, learned | Yes |
| GLA | 5.0399 ± 0.012 | 5.0245 | 5.0526 | 5.0426 | Sc, learned | No |
| ScalarStatic | 5.0884 ± 0.004 | 5.0936 | 5.0846 | 5.0871 | Sc, static | No |
| Standard Attention | 5.0933 ± 0.006 | 5.1009 | 5.0912 | 5.0877 | – (softmax) | – |
| StaticChannel | 5.1505 ± 0.010 | 5.1572 | 5.1367 | 5.1575 | Ch, static | No |

**Takeaway.** The same top-four delta-rule grouping appears on WikiText-103, indicating that the main delta-vs-no-delta separation is not specific to TinyStories.

## 4.4 Cross-Dataset Comparison

Table 6 compares validation loss and rank across both datasets. The top-4 order is identical, and only minor rank swaps occur in the non-delta group.

Table 6: Cross-dataset comparison showing rank stability. Top-4 order is identical on both datasets.

| Model | TinyStories | Rank | WikiText-103 | Rank | Gate | Delta? |
|---|---|---|---|---|---|---|
| **StaticChannelDelta** | **2.2591** | **1** | **4.9207** | 1 | Ch, static | Yes |
| KDA | 2.2820 | 2 | 4.9353 | 2 | Ch, learned | Yes |
| ScalarStaticDelta | 2.3015 | 3 | 4.9704 | 3 | Sc, static | Yes |
| DeltaNet | 2.3222 | 4 | 4.9869 | 4 | Sc, learned | Yes |
| Standard | 2.3833 | 5 | 5.0933 | 7 | – | – |
| GLA | 2.4207 | 6 | 5.0399 | 5 | Sc, learned | No |
| ScalarStatic | 2.4307 | 7 | 5.0884 | 6 | Sc, static | No |
| StaticChannel | 2.5491 | 8 | 5.1505 | 8 | Ch, static | No |

The ranking is remarkably stable across both datasets and all 3 seeds (Figure 2). The top-4 positions are occupied by the same **four delta-rule variants** in every seed×dataset cell: StaticChannelDelta, KDA, ScalarStaticDelta, and DeltaNet. The ordering within the top 4 varies between seeds (e.g., KDA takes rank 1 on TinyStories seed 7), but no non-delta variant ever enters the top 4. The bottom position (StaticChannel, rank 8) is also consistent. The only rank swaps occur among the non-delta variants (ranks 5–7), where Standard, GLA, and ScalarStatic reorder slightly between datasets.

At 18M, StaticChannelDelta holds rank 1 in 5 of 6 dataset×seed combinations (KDA takes rank 1 on TinyStories seed 7). However, the margins among the top-3 are small relative to seed variance (e.g., the 0.023 TinyStories gap between StaticChannelDelta and KDA is comparable to KDA's cross-seed std of 0.020), and with only 3 seeds per condition, formal significance tests are underpowered. We therefore emphasize the robust *structural* findings (the delta vs. non-delta divide and the interaction effect) over the specific ranking within the top-3, which is scale-dependent (Section 4.10).

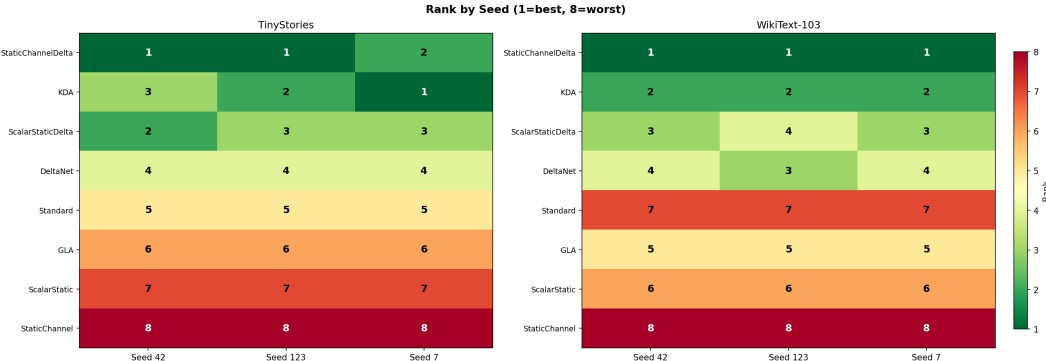

Figure 2: Rank by seed (1=best, 8=worst) across 3 seeds and 2 datasets. The same four delta-rule variants occupy the top 4 in every column; their internal ordering varies by seed.

A clear factorial pattern emerges. Among the four data-independent variants, we can isolate the effects of granularity and the delta rule (Table 7):

Table 7: TinyStories factorial decomposition (mean val loss). Channel-wise granularity hurts without the delta rule ($+0.118$) but helps with it ($-0.042$), revealing an interaction effect.

|  | No Delta Rule | Delta Rule | $\Delta$ (delta effect) |
|---|---|---|---|
| **Scalar** | ScalarStatic (2.4307) | ScalarStaticDelta (2.3015) | $-0.129$ |
| **Channel-wise** | StaticChannel (2.5491) | StaticChannelDelta (2.2591) | $-0.290$ |
| $\Delta$ **(channel effect)** | $+0.118$ (worse) | $-0.042$ (better) |  |

The same pattern holds on WikiText-103 (Table 8):

Table 8: WikiText-103 factorial decomposition. The interaction replicates: channel-wise hurts without delta ($+0.062$) but helps with it ($-0.050$).

|  | No Delta Rule | Delta Rule | $\Delta$ (delta effect) |
|---|---|---|---|
| **Scalar** | ScalarStatic (5.0884) | ScalarStaticDelta (4.9704) | $-0.118$ |
| **Channel-wise** | StaticChannel (5.1505) | StaticChannelDelta (4.9207) | $-0.230$ |
| $\Delta$ **(channel effect)** | $+0.062$ (worse) | $-0.050$ (better) |  |

This interaction is the key finding (see also Figure 3 for a visual summary): channel-wise decay without the delta rule introduces *more interference* (each dimension accumulates superimposed associations at its own rate, with no mechanism to correct errors), while channel-wise decay *with* the delta rule enables multi-timescale corrective memory. This is consistent with the delta rule being especially beneficial when multiple decay timescales are present. The effect replicates across both datasets, with the delta rule's benefit consistently larger for channel-wise ($-0.290$ / $-0.230$) than scalar ($-0.129$ / $-0.118$) gating. The interaction magnitude is substantial: on TinyStories, the differential delta benefit (channel-wise minus scalar: $0.290 - 0.129 = 0.161$) is 77% of the average delta main effect (0.210), indicating that the interaction is nearly as large as the primary factor.

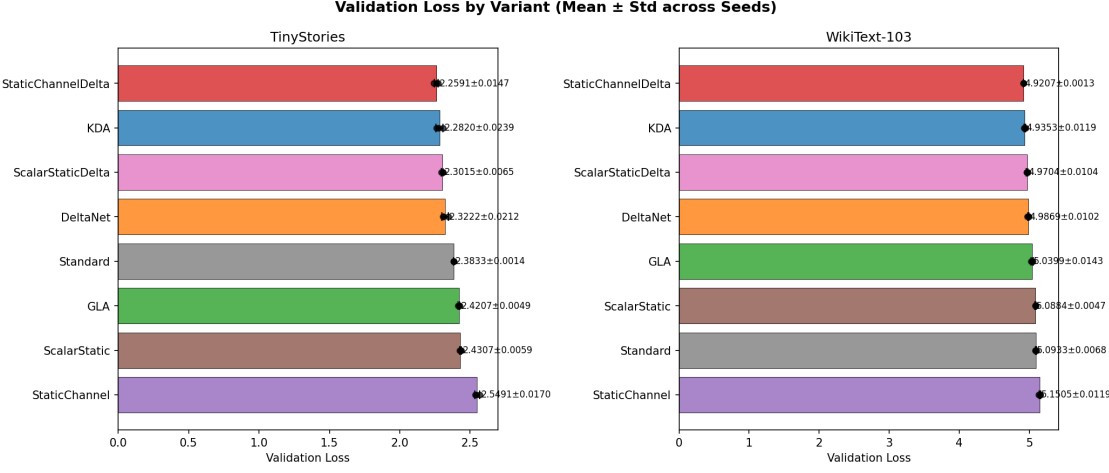

Figure 3: Validation loss by variant (mean ± std across seeds). Black dots show individual seed results.

## 4.5 Training Curves

Figure 4 shows validation loss vs. training step for all variants on both datasets. The delta vs. non-delta separation is already visible at the first epoch-level validation checkpoint and is maintained through the fixed training budget. However, all curves are still improving at the final checkpoint, so these trajectories should be interpreted as fixed-budget comparisons rather than evidence of convergence-optimal rankings.

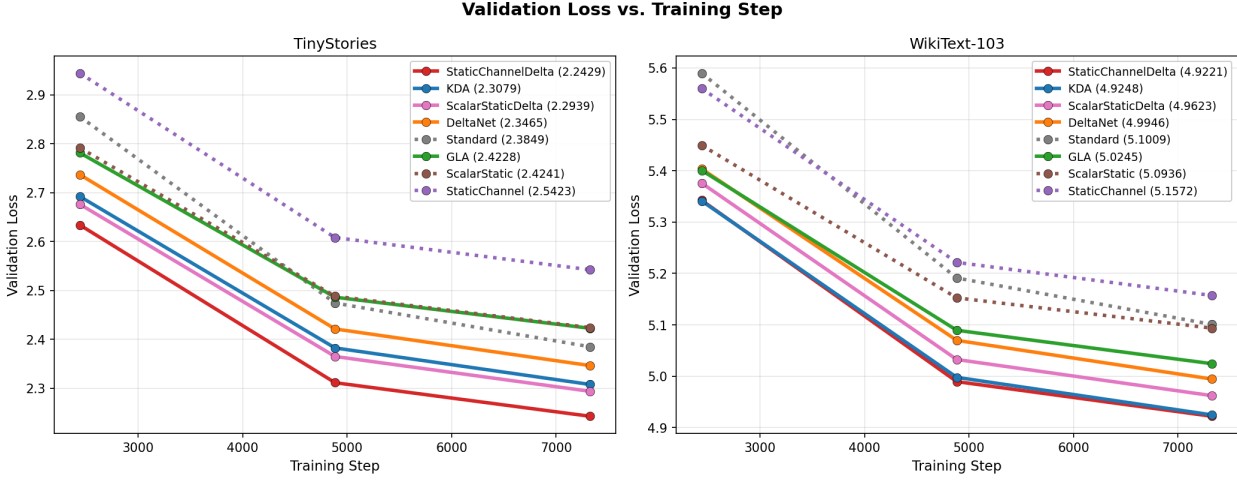

Figure 4: Validation loss vs. training step on TinyStories (left) and WikiText-103 (right). Final val loss shown in legend.

## 4.6 Key Findings

**Finding 1: The same four delta-rule variants occupy the top 4 in every seed×dataset cell.** The *mean* ranking is StaticChannelDelta > KDA > ScalarStaticDelta > DeltaNet on both datasets, but the internal ordering varies by seed (e.g., KDA takes rank 1 on TinyStories seed 7). No non-delta variant ever enters the top 4, and no delta variant ever falls below rank 4. The bottom position (StaticChannel, rank 8) is also consistent.

**Finding 2: The delta/non-delta split is visible throughout the logged validation checkpoints.** The training curves (Figure 4) show that the main structural separation is present from the first epoch-level validation checkpoint onward. Because the curves are still descending at the final checkpoint, we do not interpret this as a convergence result.

**Finding 3: StaticChannelDelta is the best mean performer at 18M on both datasets**, achieving rank 1 in 5 of 6 dataset×seed conditions (KDA takes rank 1 on TinyStories seed 7). On TinyStories, the mean margin over KDA is 0.023 (2.2591 vs 2.2820). On WikiText-103, the mean margin is 0.015 (4.9207 vs 4.9353). Although the WikiText-103 margin is small in absolute terms, it is consistent: StaticChannelDelta beats KDA in every individual seed (4.9221 vs 4.9248, 4.9195 vs 4.9482, 4.9206 vs 4.9328), with StaticChannelDelta's cross-seed standard deviation (0.001) an order of magnitude smaller than KDA's (0.010). On WikiText-103, StaticChannelDelta has the lowest cross-seed variance of any variant (0.001); on TinyStories, several non-delta variants have lower variance (e.g., Standard 0.001, GLA 0.004). However, at 125M parameters, KDA overtakes StaticChannelDelta (Section 4.10), indicating that this advantage is scale-dependent.

**Finding 4: The delta rule is the strongest observed factor in our controlled fixed-budget comparisons.** All four delta-rule variants (ranks 1–4) beat every model without a delta-rule write, including the softmax baseline, on both datasets (Figure 3). The gap is large: on TinyStories, the worst delta variant (DeltaNet, 2.3222) beats the best model without a delta-rule write (Standard, 2.3833) by 0.061. In the complete data-independent $2 \times 2$ comparison, the delta rule reduces loss by 0.129 (scalar) to 0.290 (channel-wise). Averaged over these matched data-independent pairs, the delta-rule improvement is 0.210 nats on TinyStories and 0.174 on WikiText-103, compared with average absolute granularity effects of 0.080 and 0.056. Across the matched conditioning comparisons available in our design (scalar non-delta, scalar delta, and channel-wise delta), the average absolute static-vs-data-dependent gap is smaller: 0.018 nats on TinyStories and 0.027 on WikiText-103. We therefore use "strongest observed factor" to refer to these matched loss gaps and rank separations, not to a formal variance decomposition.

**Finding 5: In the studied LM settings, channel-wise granularity helps only with the delta rule (interaction effect).** Without the delta rule, channel-wise gating *hurts*: ScalarStatic (2.4307) beats StaticChannel (2.5491) by 0.118 on TinyStories. With the delta rule, channel-wise gating *helps*: StaticChannelDelta (2.2591) beats ScalarStaticDelta (2.3015) by 0.042. Among data-dependent variants, the same pattern holds for the available scalar-vs-channel delta comparison: KDA (channel-wise + delta) beats DeltaNet (scalar + delta). This interaction is consistent with channel-wise decay being most effective when paired with a corrective write mechanism.

**Finding 6: Data-independent gating shows a small edge at 18M that reverses at 125M.** At 18M, StaticChannelDelta beats KDA on both datasets (gaps: 0.023 TinyStories, 0.015 WikiText-103) and ScalarStaticDelta beats DeltaNet (gaps: 0.021 TinyStories, 0.017 WikiText-103). However, the advantage is modest compared to the delta rule effect, and at 125M parameters KDA overtakes StaticChannelDelta (Section 4.10), consistent with data-dependent gating benefiting more from increased capacity.

**Finding 7: Delta-rule linear attention variants outperform the softmax baseline in this setup.** In our controlled experiments, the top-4 delta-rule variants all achieve lower validation loss than standard softmax attention on both datasets. On WikiText-103, all linear attention variants except StaticChannel beat softmax. On TinyStories, standard attention (rank 5) beats the linear-attention variants without delta-rule writes (ranks 6–8), suggesting that without the delta rule, linear attention's memory is not necessarily superior to softmax at this scale.

## 4.7 Analysis: Learned Decay Spectrum and State Energy

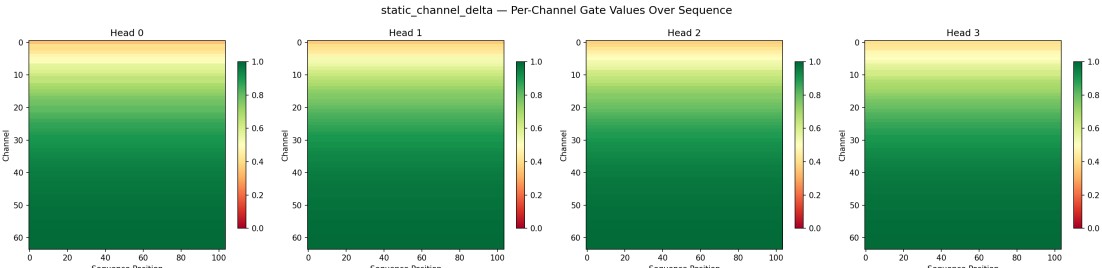

Figure 5: StaticChannelDelta per-channel gate values across sequence positions (4 heads). Horizontal bands confirm data-independent gating. Low-index channels (indices 0–10) have gates near 0.37 (fast forgetting); high-index channels (indices 50–63) have gates near 1.0 (slow memory), matching the exponential spectrum initialization (Section 3.3). KDA comparison in Appendix D.

StaticChannelDelta's exponential spectrum initialization is largely preserved after training, creating a fixed multi-timescale structure (Figure 5). High-index channels serve as "slow memory lanes" retaining information over hundreds of tokens, while low-index channels track only recent context. This produces stable state energy distributions: energy concentrates in slow-decay channels, with smooth trajectories across the sequence (see Figure 9 in Appendix D). In contrast, KDA's data-dependent gates fluctuate per token (Figures 8 and 10), producing more volatile energy patterns that may contribute to its higher cross-seed variance.

## 4.8 Synthetic Associative Recall

To test the multi-timescale memory hypothesis beyond perplexity, we evaluate all eight model variants on a synthetic associative recall task. We present this as a probe of mechanism properties rather than a comprehensive benchmark; it tests one specific capability (exact key-value retrieval) under controlled conditions (single seed, single task). The task presents 4 key-value pairs drawn from disjoint token sets (64 keys, 64 values), followed by a variable number of distractor tokens, then queries a random key; the model must predict the associated value. The *retrieval distance d* is the number of distractor tokens between the last key-value pair and the query; at $d=0$ the query immediately follows the pairs, while $d=512$ inserts 512 irrelevant tokens that the model must retain information through. Accuracy is the fraction of test sequences where the model's highest-probability prediction at the answer position matches the correct value token. With 64 possible values, *chance-level* accuracy is $1/64 \approx 1.6\%$. We train a fresh tiny model (5.1M params, compact 256-token vocab) per variant for 15,000 steps with loss computed **only at the answer position** (the last token), and evaluate at 8 retrieval distances (0–512 distractor tokens).

Table 9: Associative recall accuracy by retrieval distance (number of distractor tokens). Chance level: $1/64 = 0.016$.

| Variant | d=0 | d=16 | d=32 | d=64 | d=128 | d=256 | d=384 | d=512 | Mean |
|---|---|---|---|---|---|---|---|---|---|
| **DeltaNet** | **1.000** | **1.000** | **1.000** | **1.000** | **1.000** | **1.000** | **1.000** | **0.990** | **0.999** |
| **ScalarStaticDelta** | **1.000** | **1.000** | **1.000** | **1.000** | **1.000** | **1.000** | **1.000** | 0.342 | **0.918** |
| StaticChannel | 0.297 | 0.271 | 0.301 | 0.258 | 0.271 | 0.260 | 0.266 | 0.279 | 0.275 |
| StaticChannelDelta | 0.273 | 0.244 | 0.299 | 0.248 | 0.285 | 0.270 | 0.289 | 0.295 | 0.275 |
| ScalarStatic | 0.307 | 0.258 | 0.291 | 0.244 | 0.291 | 0.234 | 0.287 | 0.256 | 0.271 |
| GLA | 0.281 | 0.262 | 0.270 | 0.250 | 0.262 | 0.268 | 0.271 | 0.266 | 0.266 |
| KDA | 0.283 | 0.270 | 0.250 | 0.271 | 0.248 | 0.291 | 0.260 | 0.252 | 0.266 |
| Standard | 0.254 | 0.238 | 0.275 | 0.252 | 0.270 | 0.279 | 0.275 | 0.260 | 0.263 |

Table 9 reports accuracy at each retrieval distance. The results reveal a striking two-tier pattern (noting that this is a single-seed probe and the pattern should be validated with additional seeds):

**Tier 1: Near-perfect under this probe (>90% mean accuracy):** DeltaNet (99.9%) and ScalarStaticDelta (91.8%). Both achieved training loss of 0.000. DeltaNet maintains perfect accuracy through $d$=384 with only a slight dip at $d$=512 (99.0%). ScalarStaticDelta is perfect through $d$=384 but drops at $d$=512 (34.2%).

**Tier 2: Partially learned (∼27% mean accuracy):** All other 6 variants, including KDA, StaticChannelDelta, GLA, StaticChannel, ScalarStatic, and Standard. These achieved final training losses of ∼1.4 and show flat accuracy across distances with no distance-dependent degradation.

The grouping is **not** the same as for language modeling. On LM perplexity, the decisive axis was delta vs. non-delta (all delta variants ranked top 4). On associative recall, the decisive axis is **scalar decay + delta rule** (Table 10):

Table 10: Associative recall grouped by granularity and delta-rule use. Entries report mean accuracy over retrieval distances from Table 9.

|  | **No Delta Rule** | **Delta Rule** |
|---|---|---|
| **Scalar gate** | GLA (26.6%), ScalarStatic (27.1%) | DeltaNet (**99.9%**), ScalarStaticDelta (**91.8%**) |
| **Channel-wise gate** | StaticChannel (27.5%) | KDA (26.6%), StaticChannelDelta (27.5%) |

Under this probe and training budget, the delta rule is necessary for high accuracy (no non-delta variant reaches the top tier), but not sufficient; it must be paired with scalar decay. Channel-wise decay appears to interfere with the delta rule's exact retrieval mechanism on this task: by applying different forgetting rates to different dimensions of the same key-value pair, channel-wise gating may fragment the stored association, preventing exact recall even with corrective updates. A visualization of the same data is provided in Appendix B.

**Budget sweep: the plateau is not resolved by doubling the budget.** To test whether channel-wise delta variants are merely slow to converge, we reran KDA and StaticChannelDelta with 2× the training budget (30K steps). Neither improved meaningfully (Table 11):

Table 11: Budget sweep for channel-wise delta variants on associative recall.

| Variant | 15K steps | 30K steps | Δ |
|---|---|---|---|
| KDA | 0.266 | 0.277 | +0.011 |
| StaticChannelDelta | 0.275 | 0.282 | +0.007 |
| (DeltaNet, 15K) | 0.999 | – | – |

Both variants gained ∼1 percentage point, within noise, while remaining at training loss ∼1.4. This is consistent with, but does not prove, a structural difficulty rather than a pure convergence-speed difference: channel-wise decay may fragment stored key-value bindings across dimensions with different decay rates, making it harder for the delta rule's corrective update to recover coherence.

**Task-dependent tradeoff.** These results, combined with the LM perplexity results (Sections 4.2–4.4), reveal a task-dependent tradeoff. On language modeling, where the model must represent a rich distribution over continuations, channel-wise decay provides beneficial multi-timescale structure that the delta rule can leverage (StaticChannelDelta rank 1). On exact associative recall, where the model must store and retrieve specific bindings, uniform scalar decay preserves the coherence of stored pairs (DeltaNet rank 1). This suggests that channel-wise and scalar gating occupy complementary niches in the design space, and that the optimal choice depends on whether the task rewards representational richness or retrieval precision.

### 4.9   Initialization Ablation

The exponential spectrum initialization (Section 3.3) is a core design choice of StaticChannelDelta. To test whether the specific initialization matters, or whether any learnable decay rates suffice, we train StaticChannelDelta on TinyStories with three initialization strategies (results in Table 12):

- **Exponential** (default): $\gamma_i = \exp(-2^{-8i/d})$, producing a structured spectrum from fast ($\gamma_0 \approx 0.37$) to slow ($\gamma_{d-1} \approx 0.995$) decay.

- **Identical**: All channels initialized to the same rate ($\gamma = 0.9$). This tests whether channel diversity at initialization matters.

- **Random**: Each channel drawn uniformly from $[0.3, 0.999]$. This provides diverse timescales but without the structured ordering of the exponential spectrum.

Table 12: Initialization ablation for StaticChannelDelta on TinyStories.

| Initialization | Val Loss | Δ vs Exponential |
|---|---|---|
| **Exponential** | **2.2429** | – |
| Random | 2.2767 | +0.034 |
| Identical | 2.3681 | +0.125 |

The exponential spectrum is the best initialization, and the ranking is informative:

**Identical initialization is worst** (+0.125). Starting all channels at the same rate effectively removes the multi-timescale structure that makes channel-wise gating valuable. The resulting model performs close to ScalarStaticDelta (2.3015), confirming that channel diversity, not just having per-channel parameters, drives the benefit.

**Random is intermediate** (+0.034). The model can learn reasonable timescales from an unstructured starting point, but the exponential spectrum provides a better inductive bias. The gap suggests that the smooth, ordered progression from fast to slow channels is preferable to arbitrary initial rates.

All three initializations are learnable: the gates can move freely during training. The fact that the structured exponential init still wins suggests that the optimization landscape is easier to navigate when starting from a well-ordered timescale spectrum.

### 4.10   Scale-Up: 42M and 125M Parameters

To test whether our findings hold beyond 18M parameters, we train at two additional scales on TinyStories (Table 13). At $\sim$42M ($d_{\mathrm{model}} = 512$, 8 layers, 8 heads), we train the top 4 delta-rule variants and standard attention (single seed). At $\sim$125M ($d_{\mathrm{model}} = 768$, 12 layers, 12 heads), we train all eight model variants with 3 seeds (42, 123, 7), matching the 18M protocol. Because the 125M experiment is run only on TinyStories, we interpret the KDA-vs-StaticChannelDelta reversal as a TinyStories scale-up result rather than a cross-dataset scaling conclusion.

Table 13: Scale-up on TinyStories. 18M and 125M columns show mean $\pm$ std across 3 seeds; 42M is single seed. The delta vs. non-delta divide persists across scales; within-delta rankings shift.

| Model | 18M (3-seed) | R | 42M | R | 125M (3-seed) | R |
|---|---|---|---|---|---|---|
| KDA | $2.2820 \pm 0.020$ | 2 | 1.9703 | 2 | $\mathbf{1.8614 \pm 0.002}$ | **1** |
| StaticChannelDelta | $\mathbf{2.2591 \pm 0.012}$ | **1** | **1.9695** | **1** | $1.8687 \pm 0.006$ | 2 |
| ScalarStaticDelta | $2.3015 \pm 0.005$ | 3 | 1.9844 | 3 | $1.8705 \pm 0.005$ | 3 |
| DeltaNet | $2.3222 \pm 0.017$ | 4 | 2.0176 | 4 | $1.9004 \pm 0.006$ | 4 |
| Standard | $2.3833 \pm 0.001$ | 5 | 2.0735 | 5 | $1.9570 \pm 0.008$ | 5 |
| GLA | $2.4207 \pm 0.004$ | 6 | – | – | $1.9710 \pm 0.007$ | 6 |
| ScalarStatic | $2.4307 \pm 0.005$ | 7 | – | – | $2.0075 \pm 0.008$ | 7 |
| StaticChannel | $2.5491 \pm 0.014$ | 8 | – | – | $2.0079 \pm 0.005$ | 8 |

**KDA overtakes StaticChannelDelta at 125M.** Defining the gap as (KDA loss $-$ SCD loss), so that positive values indicate SCD wins, the gap evolves from $+0.023$ (18M) to $+0.0008$ (42M) to $-0.007$ (125M). KDA wins 2 of 3 seeds at 125M; StaticChannelDelta wins 1. This is consistent with data-dependent gating benefiting disproportionately from increased capacity. Notably, the stability advantage reverses: KDA's cross-seed std at 125M (0.002) is 3$\times$ lower than StaticChannelDelta's (0.006), suggesting that data-dependent gates optimize more stably at this scale.

**The top-3 compress to near-indistinguishable.** The spread among the top 3 shrinks from 0.042 (18M) to 0.015 (42M) to just **0.009** (125M). At 125M, the gap between ranks 1 and 3 (0.009) is comparable to individual seed variance ($\sim$0.005), suggesting that at this scale, the presence of the delta rule matters more than the specific gating choice among the strongest delta-rule variants.

**The delta-rule advantage is the dominant observed effect at every tested scale.** At 125M, the gap between the worst delta variant (DeltaNet, $1.900 \pm 0.006$) and the best model without a delta-rule write (Standard, $1.957 \pm 0.008$) is **0.057**, 6$\times$ larger than the top-3 spread. The delta-vs-standard gap is stable across all three scales (0.061 at 18M, 0.056 at 42M, 0.057 at 125M). The top 4 = all delta, bottom 4 = no delta-rule write split holds at 18M (both datasets) and 125M (TinyStories) where all eight model variants were run; at 42M (5 variants), all four delta variants beat Standard. The bottom 4 ranking (Standard, GLA, ScalarStatic, StaticChannel) is unchanged from 18M to 125M under this fixed-budget protocol.

### 4.11 Exploratory Longer Sequence Length: seq_len=1024

A natural concern is whether seq_len=512 is too short for decay mechanisms to matter, and that perhaps all variants look similar because there is little long-range context to forget. As an exploratory stress test, we repeat the top 4 delta-rule variants and standard attention on TinyStories at seq_len=1024 (same 10M token budget, same hyperparameters, single seed). This experiment is not part of the primary claims because it is single-seed and includes only 5 of 8 model variants. Table 14 compares the results at both sequence lengths.

Table 14: Longer sequence length experiment on TinyStories (single seed).

| Model | Val Loss (1024) | Val Loss (512) | Rank (1024) | Rank (512) |
|---|---|---|---|---|
| **StaticChannelDelta** | **2.4214** | **2.2429** | **1** | **1** |
| ScalarStaticDelta | 2.4409 | 2.2939 | 2 | 2 |
| KDA | 2.4457 | 2.3079 | 3 | 3 |
| DeltaNet | 2.4476 | 2.3465 | 4 | 4 |
| Standard | 2.6358 | 2.3849 | 5 | 5 |

Raw losses are higher at 1024 because the same 10M token budget yields fewer, longer sequences (9,765 samples vs. $\sim$19,500 at 512), reducing effective training data. The key comparisons are relative:

**The gap between delta variants and standard attention widens.** At 512, StaticChannelDelta beats Standard by 0.142; at 1024, this grows to **0.214**. This provides preliminary evidence that decay mechanisms may become more advantageous at longer sequences, though the result should be interpreted cautiously because the experiment is single-seed.

**Delta variants compress.** The spread among the top 4 shrinks from 0.104 (at 512) to **0.026** (at 1024), consistent with the delta rule itself becoming the dominant factor at longer sequences, though this experiment is single-seed and should be interpreted cautiously.

**StaticChannelDelta remains rank 1** at 18M with seq_len=1024, consistent with the multi-seed 18M results (Sections 4.2–4.3), though this comparison uses a single seed and only 5 of 8 variants.

## 5 Discussion

### 5.1 Static vs. Data-Dependent: Scale-Dependent

At 18M parameters with 3-seed validation, StaticChannelDelta beats KDA in mean loss on both datasets. The advantage is 0.023 on TinyStories and 0.015 on WikiText-103, though KDA wins one individual seed (TinyStories seed 7). However, the advantage is scale-dependent: the gap narrows from 0.023 (18M) to 0.0008 (42M) and *reverses* at 125M, where KDA wins by 0.007 across 3 seeds (Section 4.10). KDA wins 2 of 3 seeds at 125M while StaticChannelDelta wins 1, indicating that the within-top-3 ranking is noisy at this scale.

We hypothesize that data-dependent gating provides two ingredients: (a) multi-timescale memory structure, and (b) content-adaptive forgetting. Static gating provides only (a). One possible explanation for the scale dependence is that the content-adaptive component is harder to use effectively at small scale, but becomes more beneficial as model capacity increases and the model can learn genuinely useful input-dependent decay patterns. The stability picture also reverses: at 18M on WikiText-103, StaticChannelDelta has the lowest cross-seed variance of any variant (std=0.001 vs. KDA's 0.010), but at 125M, KDA's variance (0.002) is $3\times$ lower than StaticChannelDelta's (0.006).

The practical advantages of StaticChannelDelta remain relevant in some settings:

1. **Near-identical perplexity.** At 125M, the 3-seed gap is 0.007 nats ($1.869 \pm 0.006$ vs. $1.861 \pm 0.002$). The top-3 spread (0.009) is comparable to individual seed variance, making the ranking within this group effectively a tie.

2. **Fewer parameters.** Static gates are $H \times d$ learnable scalars; KDA requires a full projection layer to produce gates from input, adding $\sim$7M parameters at the 125M scale (131M vs. 124M).

3. **Simpler optimization.** Static gates decouple the timescale structure (learned once during training) from content-dependent decisions (handled by keys, values, and the delta rule).

4. **Stability.** HALO reports that KDA failed to converge during distillation (Appendix G.2), while Lightning Attention (data-independent scalar gates) was stable. Our StaticChannelDelta extends this stability advantage to channel-wise granularity. A preliminary NoPE-based recurrence stability test (Appendix C) confirms that all tested linear attention variants are stable at evaluation lengths up to $4\times$ training length.

### 5.2 Computational Advantages of Static Gates

Beyond perplexity, data-independent gates offer a practical advantage for efficient implementation. In chunkwise parallel training (Yang et al., 2023), the sequence is divided into chunks of length $C$, and within each chunk the intra-chunk attention is computed in parallel while inter-chunk state propagation uses the recurrence. For data-dependent gates (GLA, KDA), the gate values must be *computed from the input* before the decay matrix for each chunk can be formed, creating a sequential dependency between the gate projection and the chunk computation. For data-independent gates (StaticChannelDelta, ScalarStatic), the

decay matrix $\mathbf{\Gamma}^C$ (the gate raised to the $C$-th power) can be **precomputed once** and reused across all chunks and all sequences, eliminating this dependency. This simplification is particularly significant for channel-wise gates: KDA requires a per-token $d$-dimensional gate vector ($B \times H \times T \times d$ values), while StaticChannelDelta needs only $H \times d$ fixed values for the entire forward pass. We note that HALO (Chen et al., 2026) similarly observes that Lightning Attention's fixed scalar decay simplifies kernel implementation; our StaticChannelDelta extends this advantage to the channel-wise setting. We emphasize that this is an implementation-level argument about kernel structure and dependency patterns rather than a speed benchmark measured in this paper.

## 5.3 Task-Dependent Tradeoff: Language Modeling vs. Exact Recall

Our associative recall experiment (Section 4.8) reveals that the best decay mechanism depends on the task. On language modeling, StaticChannelDelta (channel-wise + delta) is the best mean performer at 18M on both datasets. On exact recall, DeltaNet (scalar + delta) achieves 99.9% accuracy while StaticChannelDelta plateaus at 27.5%.

One possible interpretation is that this reflects a task-dependent tradeoff between two desirable properties of the recurrent state:

- **Multi-timescale richness.** Language modeling requires representing a distribution over continuations conditioned on diverse contextual cues at different temporal scales: recent syntax, medium-range topic, long-range facts. Channel-wise decay enables different state dimensions to specialize at different timescales, and the delta rule manages the resulting interference. This combination produces the best LM perplexity.

- **Retrieval precision.** Exact associative recall requires storing and retrieving specific key-value bindings without corruption. When all dimensions decay at the same rate (scalar gating), a stored association $\mathbf{k} \otimes \mathbf{v}$ remains coherent, since all components decay together. Channel-wise decay fragments this coherence: different dimensions of the same binding decay at different rates, distorting the stored association and making exact retrieval harder even with the delta rule's corrective update.

These results suggest a possible task-dependent tradeoff: **decay granularity may trade off representation richness against binding precision, with the delta rule necessary for both.** Gate granularity selects the operating point on this spectrum:

$$\begin{array}{ccc}
\text{Representation richness} & \longleftrightarrow & \text{Retrieval precision} \\
\text{Channel-wise + delta} & & \text{Scalar + delta} \\
\text{(best LM perplexity)} & & \text{(best exact recall)}
\end{array}$$

This tradeoff suggests that the optimal gating strategy is task-dependent. Hybrid architectures that combine scalar-gated layers (for precise retrieval) with channel-gated layers (for rich representation) across different layers of the same model could potentially capture both properties.

A training budget sweep (Section 4.8) provides evidence that the recall plateau is not merely a short-budget artifact: doubling the training budget from 15K to 30K steps yielded no meaningful improvement for channel-wise delta variants (KDA: +0.011, StaticChannelDelta: +0.007), while both remained near $\sim$28% accuracy.

## 5.4 Implications for Model Design

The practical implication is not that a single decay gate should be used everywhere, but that the delta-rule write should be treated as the default choice in this design space. For language modeling under a matched training budget, the strongest group is channel-wise+delta: KDA and StaticChannelDelta occupy the top two mean ranks at 18M on both datasets, and remain in the top two at 125M on TinyStories. For exact key-value binding, the recall probe points in the opposite direction: scalar+delta variants preserve binding coherence and achieve the highest retrieval accuracy. This suggests a concrete hybrid design direction for

existing linear-attention systems: use delta-rule layers throughout, but mix scalar-gated layers where precise retrieval is important with channel-wise-gated layers where multi-timescale representation is useful. This conclusion is an architectural implication of our controlled comparisons rather than a measured hybrid benchmark.

### 5.5 Why Is the Delta Rule Essential?

Without the delta rule, the state update $\mathbf{S}_t = \boldsymbol{\Gamma} \odot \mathbf{S}_{t-1} + \phi(\mathbf{k}_t) \otimes \mathbf{v}_t$ blindly accumulates outer products. In the presence of diverse decay rates:

- Slow-decaying channels accumulate many superimposed associations, leading to interference.

- Fast-decaying channels lose information before it can be usefully retrieved.

The delta rule may mitigate both issues: it *corrects* existing associations rather than adding new ones on top, which can reduce interference in slow channels and make the brief window of fast channels more informative. Our factorial analysis (Section 4.4) quantifies this: the delta rule's benefit is *larger* with channel-wise decay $(-0.290)$ than scalar decay $(-0.129)$, because there is more interference to correct when each dimension has its own timescale.

### 5.6 Limitations

**Fixed-budget rather than convergence-based comparison.** All main experiments compare variants after the same token budget, optimizer, schedule, and number of epochs. This design isolates performance under a matched training protocol, but it does not establish that the same ordering would hold if each model were independently tuned and trained to its best validation loss. The training curves are still improving at the final logged checkpoint, so slower-converging variants could in principle close or reverse some gaps with longer training. We therefore interpret the results as evidence about fixed-budget effectiveness and interaction structure, not asymptotic optimality.

**Scale.** We validate up to 125M parameters with 3 seeds (Section 4.10). The delta-rule vs. no-delta-rule-write divide is robust across all tested scales under our fixed-budget protocol, but the top-3 delta variants are statistically indistinguishable at 125M (spread 0.009, comparable to seed variance). Whether data-dependent gating pulls further ahead at 350M+ or the top-3 converge entirely remains unknown. Qin et al. (2025) show that some trends only emerge at 1.45B parameters. Our naive recurrent loop implementation precludes efficient training beyond 125M; validating at 350M+ would require chunkwise parallel kernels.

**Architecture scope.** All experiments use a GPT-style decoder backbone and vary only the attention mechanism. We do not test alternative backbone families, hybrid softmax/linear layer placements, or architecture-specific tuning; such choices may interact with decay-design conclusions.

**Number of seeds.** We use 3 seeds per condition at both 18M and 125M. While the structural findings (delta-rule vs. no-delta-rule-write divide, interaction effect) are consistent across all seeds, the within-top-3 ranking is noisy: KDA wins 2/3 seeds at 125M, SCD wins 1/3. 5+ seeds would enable formal statistical significance testing (e.g., paired $t$-tests).

**Sequence length.** Our main experiments use seq_len=512, with a validation at 1024 (Section 4.11) providing preliminary evidence that the advantage of decay mechanisms grows with length. However, truly long-range dependencies (4K–128K) remain untested. Our NoPE-based recurrence stability test (Appendix C) does not address HALO's hypothesis about data-dependent gates, which requires positional encodings that create length-distribution shift. A proper comparison would use RoPE or ALiBi at training length 2K–4K with evaluation at 16K–128K.

**Naive implementation.** Our recurrent loop implementation is $O(T \cdot d^2)$ per head per layer. For practical use at scale, a chunkwise parallel kernel (as in FlashLinearAttention) would be needed.

**Limited downstream tasks.** Beyond perplexity, we evaluate only on a synthetic associative recall task (Section 4.8). The empirical scope is language modeling plus one controlled retrieval probe; we do not

make claims about multimodal settings, instruction tuning, or downstream task transfer. More diverse tasks such as needle-in-haystack, MRCR, and in-context learning would further test the multi-timescale memory hypothesis. The recall experiment uses a single seed; multi-seed recall runs would strengthen the finding.

## 6 Conclusion

By evaluating seven linear-attention variants plus a softmax baseline across two datasets with 3 random seeds at 18M and scaling to 125M on TinyStories (3 seeds), we characterize decay-mechanism interactions within a shared implementation framework. The complete factorial decomposition is over granularity and delta rule in the data-independent subspace, supplemented by controlled data-dependent pairwise comparisons:

1. **The delta rule is the strongest observed factor in our controlled fixed-budget comparisons.** All four delta-rule variants (ranks 1–4) outperform every model without a delta-rule write (ranks 5–8, including the softmax baseline) on both datasets at 18M and on TinyStories at 125M, with the delta advantage also confirmed at 42M (5 variants). This divide is larger than the top-3 delta spread and larger than the granularity effects measured in the complete data-independent $2 \times 2$ subspace; pairwise static-vs-data-dependent differences among delta variants are smaller and scale-dependent.

2. **In the studied LM settings, channel-wise granularity helps only with the delta rule.** A factorial decomposition of the data-independent variants reveals a crossover interaction: without the delta rule, channel-wise decay *increases* interference (ScalarStatic beats StaticChannel); with the delta rule, channel-wise decay enables multi-timescale corrective memory (StaticChannelDelta beats ScalarStaticDelta). This is consistent with the delta rule being especially effective at leveraging diverse decay rates.

3. **Within-delta rankings are scale-dependent.** At 18M, data-independent StaticChannelDelta is the best mean performer on both datasets. At 125M, data-dependent KDA overtakes it, though the top-3 spread compresses to just 0.009 nats while the gap between the worst delta variant and the best model without a delta-rule write remains 0.057 nats.

An exploratory single-seed experiment at seq_len=1024 provides preliminary evidence that the advantage of decay mechanisms over softmax may grow with sequence length, while delta-rule variants compress. This result is not used as primary evidence because it includes only 5 of 8 model variants.

A synthetic associative recall benchmark suggests that gating granularity also creates a task-dependent tradeoff: scalar+delta variants (DeltaNet, ScalarStaticDelta) achieve above 90% mean exact key-value retrieval accuracy under the 15K-step single-seed probe, while channel-wise variants plateau at ∼27% even with doubled training budget. On this probe, channel-wise decay appears to fragment stored bindings by applying different forgetting rates to different dimensions, making exact retrieval harder. These results suggest that channel-wise gating trades retrieval precision for the multi-timescale representational richness that benefits language modeling. Hybrid architectures that combine both gating granularities across layers may be a fruitful direction.

Future work should test length generalization with positional encodings (RoPE) at longer ranges (16K–128K), scale to 350M+ parameters with chunkwise parallel kernels, and investigate whether the top-3 delta variants continue to converge or whether data-dependent gating pulls further ahead with scale.

## Reproducibility Statement

All experiments use a single codebase with shared hyperparameters across variants, ensuring that differences in results reflect only the attention mechanism under test. We report the exact model configurations (Section 4.1, Appendix A), training hyperparameters (learning rate, batch size, sequence length, optimizer settings), and random seeds (42, 123, 7) for all experiments. Each result directory contains a full training log (`log.json`) with per-step losses. We use 3 random seeds at 18M (both datasets) and 125M (TinyStories) to

quantify variance, and report all individual seed results in the appendix to enable the reader to assess effect sizes relative to seed variance. The codebase, including all training scripts, evaluation scripts, and plotting code, will be released publicly upon acceptance. All implementations are in pure PyTorch without custom CUDA kernels, ensuring broad reproducibility across hardware platforms (tested on Apple Silicon MPS and NVIDIA CUDA).

## Ethics Statement

This work is a controlled empirical study of architectural design choices in linear attention mechanisms, conducted entirely on publicly available datasets (TinyStories and WikiText-103). We do not introduce new model capabilities, train large-scale systems, or release models that could be directly misused. The findings are primarily of interest to the research community studying efficient attention mechanisms. We see no specific ethical concerns arising from this work beyond those common to language modeling research in general.

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

# A    Implementation Details

All attention variants are implemented as naive recurrent loops in PyTorch, prioritizing correctness and clarity over speed. The full codebase—including all implemented attention variants, training scripts, evaluation scripts, and plotting code—will be released publicly upon acceptance.

## A.1    Feature Maps

Non-delta variants (GLA, ScalarStatic, StaticChannel) use the ELU+1 feature map: $\phi(x) = \text{ELU}(x) + 1$, ensuring non-negative features. Delta rule variants (DeltaNet, KDA, ScalarStaticDelta, StaticChannelDelta) use $\ell_2$-normalization for keys ($\hat{\mathbf{k}}_t = \mathbf{k}_t/\|\mathbf{k}_t\|_2$) and ELU+1 for queries. The normalization is critical: without it, $\mathbf{S}^\top\mathbf{k}_t$ can grow unboundedly, causing NaN values during training.

## A.2    Model Architecture

6-layer GPT-style transformer, $d_{\text{model}} = 256$, 4 attention heads ($d_{\text{head}} = 64$), pre-norm (LayerNorm), GELU FFN with expansion factor $4\times$, learned positional embeddings (Sections 4.2–4.4) or no positional embeddings (Appendix C, NoPE mode for recurrence stability). GPT-2 tokenizer (50257 vocab). Total $\sim$18M parameters (17.7M for most variants, 18.1M for KDA due to gate projections; $\sim$17.6M in NoPE mode without positional embedding table).

## A.3    Hyperparameters

Table 15: Training hyperparameters used for the main language-modeling experiments unless otherwise specified.

| Hyperparameter | Value |
|---|---|
| Learning rate | $3 \times 10^{-4}$ |
| Warmup steps | 500 |
| LR schedule | Cosine decay |
| Weight decay | 0.1 |
| Batch size | 8 |
| Sequence length | 512 |
| Epochs | 3 |
| Gradient clip | 1.0 |
| Optimizer | AdamW ($\beta_1$=0.9, $\beta_2$=0.95) |
| Seeds | 42, 123, 7 |

# B    Associative Recall Visualization

Figure 6 visualizes the associative recall results from Table 9. We keep the table in the main text because it reports the exact values at each retrieval distance; the figure is included here only as a visual aid.

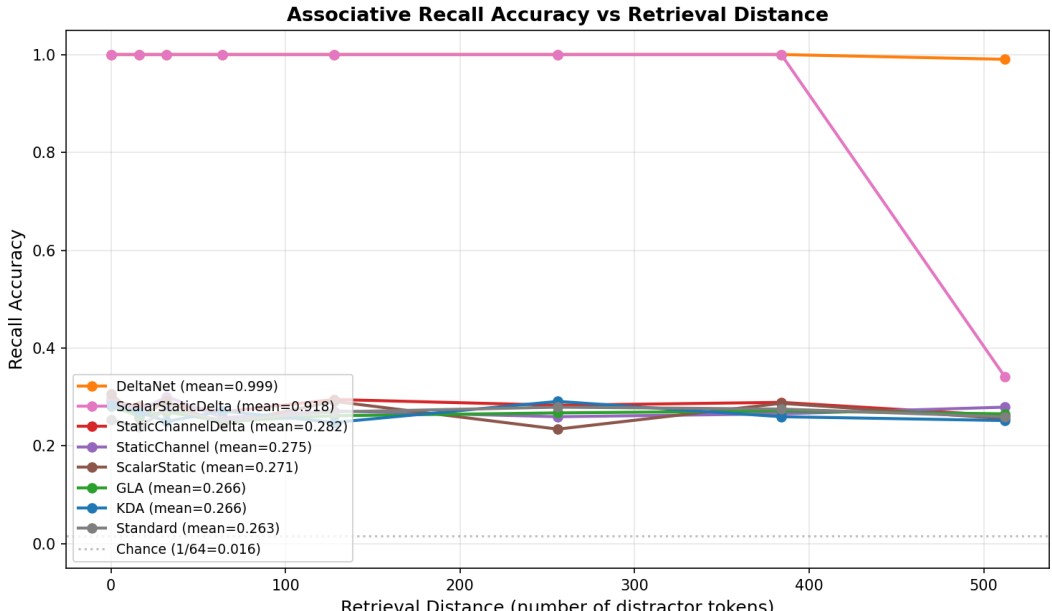

Figure 6: Associative recall accuracy vs. retrieval distance. DeltaNet and ScalarStaticDelta achieve high accuracy under the 15K-step single-seed probe; all other variants plateau near ∼27%.

## C    Recurrence Stability Under Longer Evaluation Lengths

We retrain 6 of 8 variants on WikiText-103 at sequence length 2048 without positional embeddings (NoPE mode) and evaluate at 2048, 4096, and 8192 tokens. Table 16 and Figure 7 report the results.

Table 16: Recurrence stability: validation loss at increasing evaluation lengths (trained at 2K tokens, NoPE mode).

| Model | 2K | 4K | 8K | $\Delta(2K{\rightarrow}8K)$ |
|---|---|---|---|---|
| **StaticChannelDelta** | **5.090** | **5.087** | **5.086** | **−0.004** |
| KDA | 5.099 | 5.096 | 5.094 | −0.004 |
| DeltaNet | 5.153 | 5.151 | 5.150 | −0.004 |
| GLA | 5.157 | 5.154 | 5.153 | −0.004 |
| StaticChannel | 5.318 | 5.315 | 5.314 | −0.004 |
| Standard | 5.593 | 5.622 | 5.661 | +0.068 |

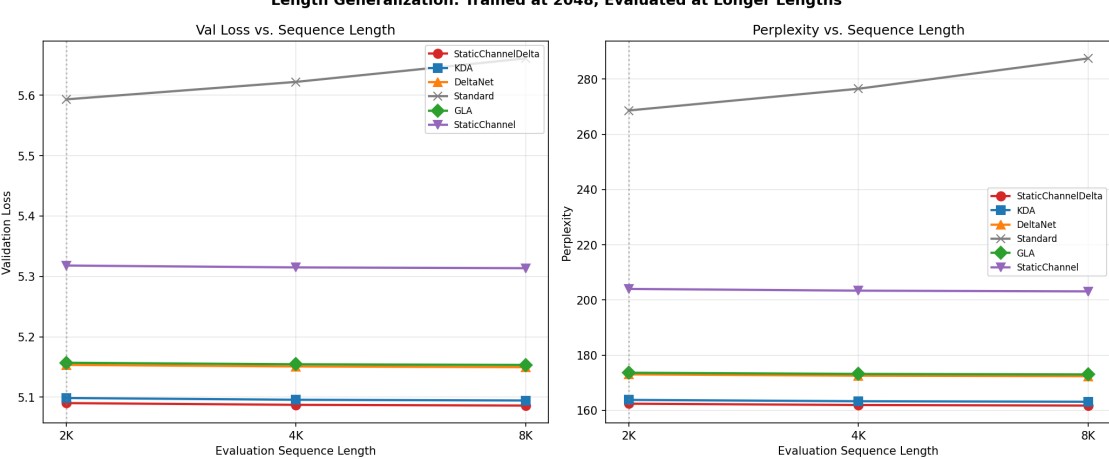

Figure 7: Val loss vs. evaluation sequence length (trained at 2K, NoPE mode). All linear attention variants improve; softmax degrades.

All linear attention variants *improve* at longer evaluation lengths ($\Delta \approx -0.004$), while standard softmax degrades ($+0.068$ at 8K). No differential advantage for static vs. data-dependent gating is observed. Note that this NoPE setup tests recurrence stability only; it does not address HALO's hypothesis about data-dependent gates, which requires positional encodings (e.g., RoPE) and longer evaluation ranges (HALO tested up to 128K).

## D Gate and State Energy Visualizations

Figure 8 shows KDA gate values, Figure 9 shows StaticChannelDelta state energy, and Figure 10 shows KDA state energy, for comparison with Figure 5 in the main text.

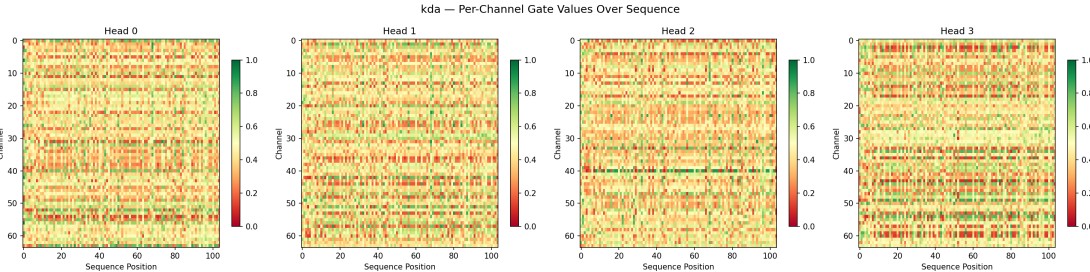

Figure 8: KDA per-channel gate values. In contrast to StaticChannelDelta's horizontal bands, KDA's data-dependent gates vary per token, producing noisy patterns.

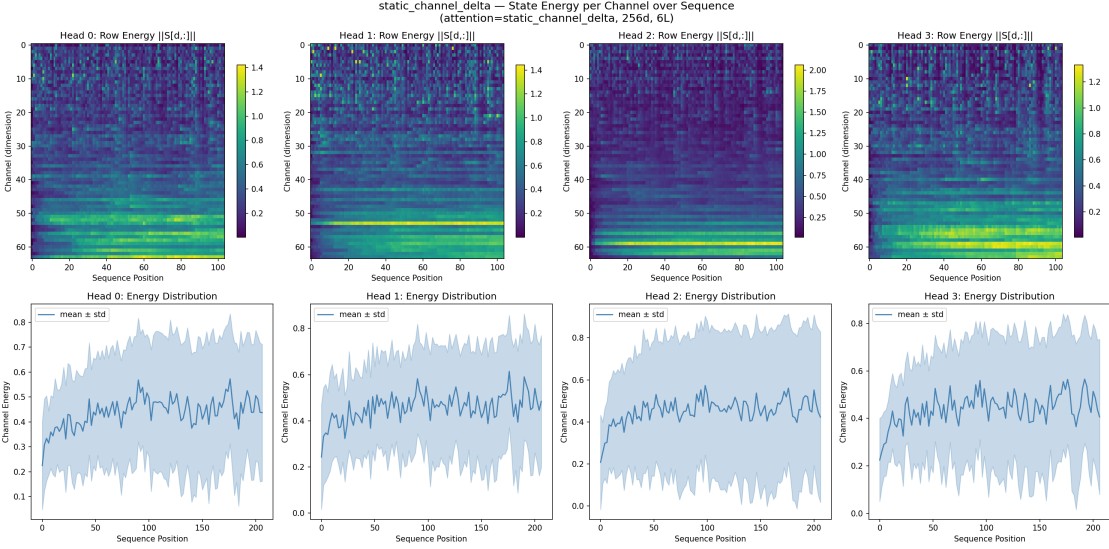

Figure 9: StaticChannelDelta state energy per channel over the sequence. Energy concentrates in slow-decay channels, with smooth trajectories.

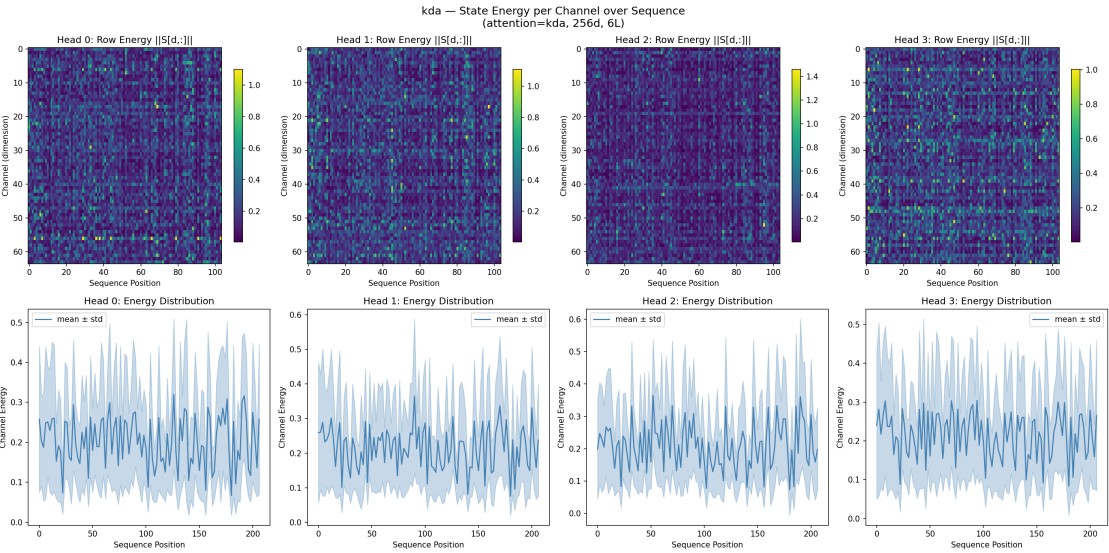

Figure 10: KDA state energy. More volatile energy patterns due to data-dependent gating, with sudden shifts when gates open or close.

# E Wall-Clock Training Times

Table 17 reports wall-clock training times. All linear attention variants use naive recurrent loop implementations (no Triton or chunkwise parallel kernels), so training times reflect the inherent per-step cost of each recurrence rather than optimized throughput. Times are measured on an Apple M4 Max (MPS backend), TinyStories, seq_len=512, batch_size=8.

Table 17: Wall-clock training times per step.

| Variant | Params (M) | ms/step | Slowdown vs Standard |
|---|---|---|---|
| Standard (softmax) | 17.7 | 20 | 1.0× |
| GLA | 17.7 | 808 | 40× |
| ScalarStatic | 17.7 | 854 | 42× |
| DeltaNet | 17.7 | 1,055 | 52× |
| ScalarStaticDelta | 17.7 | 1,187 | 59× |
| KDA | 18.1 | 1,266 | 63× |
| StaticChannel | 17.7 | 1,369 | 68× |
| **StaticChannelDelta** | 17.7 | 1,762 | 87× |

Standard attention benefits from PyTorch's optimized softmax kernel; all linear attention variants run as explicit Python loops over timesteps, explaining the large absolute gap. Within the linear attention variants, two trends are visible: (1) the delta rule adds ∼29–39% overhead (compare GLA→DeltaNet, ScalarStatic→ScalarStaticDelta, StaticChannel→StaticChannelDelta), and (2) channel-wise gating is slower than scalar gating due to element-wise decay operations. For practical deployment, chunkwise parallel kernels (as in FlashLinearAttention) would be essential; these can reduce the gap to 1.5–3× vs standard attention, as demonstrated by KDA and GLA in their respective papers.

