# OpenReview forum: "The Delta Rule Dominates: A Factorial Analysis of Decay in Linear Attention"
_TMLR — Under review for TMLR_

### Review · Reviewer_Uhc9 · 2026-06-05

**Summary Of Contributions:**

This paper studies several variants of linear attention for language modeling. The goal was to perform a controlled comparison of key design choices of "granularity", "conditioning", and "delta rule". The main finding is that, in the experimental protocol defined, the delta rule is the principal factor in determining performance ordering (using it helps). A controlled "recall" analysis also compares the designs and further supports the efficacy of the delta rule being the main factor.

## Weaknesses
- The "delta-rule" is not precisely described in the introduction. The writing in the introduction and abstract assumes the reader already knows what the delta rule is. As far as I can tell, the first time the delta rule appears is on page three. The introduction would be much clearer if the Eq (2)--(4) were presented in the introduction, and then the variants described in terms of these equations.

- The contribution claims are not precise. (1) "channel-wise granularity helps only in the presence of the
delta rule; without it, having more decay parameters actually hurts, consistent with increased interference" -- this is too broad, it need to be explicitly scoped to the tasks studied. (2) "explains more variation than granularity or conditioning at every scale": the claim should be clearer about how this is measured. (3) "task-dependent tradeoff : scalar+delta solves exact associative recall while channel-wise+delta plateau" -- ``solving'' anything is a strong claim as it implies no further work is needed and all relevant metrics are optimized, necessitating no further work on the task/area. The evidence presented doesn't support this claim. Some discussion, also, for what the

- Related to the imprecision of the claims, a major downside to the analysis is that each configuration is not fit until convergence -- the argument seems to be about the rate of improvement only, rather than the final performance. It is possible that the performance order changes once all versions reach their best validation loss. This isn't acknowledged in the limitations discussion.

- The experiment presented in Tab. 8 is the clearest controlled comparison of the different configurations, although it is subject to the same issue of potential underfitting, all versions are fit for 15k steps, so it's plausible that all variance could reach 100% accuracy if fit for longer.

- Some of the experimental analysis is not covered by the claims (e.g., longer sequence experiments)

**Additional Comments:**

- Table 1 is missing a caption, also, it's not self-contained -- the columns are not sufficiently descriptive for readers to understand the context of 'data-(in)dependent'. The abstract has a similar issue -- the exact component that may or may not be data-dependent is not clear. What constitutes data? Eq. 1 depends on previous S.

- Tab. 10: I wouldn't say "uniform" as its easily confused with "uniformly random" -- instead call it "identical" or something to distinguish it from "uniformly random"

- Fig 2. needs to be redone because the colorbar erroneously appears inside the WixiText-103 visualization.

- Table 8 and Fig 7 plot the same data. It's not clear why both are included.

**Audience:**

Yes

**Audience Explanation:**

A controlled analysis of linear attention is likely of interest to many.

**Claims And Evidence:**

No

**Claims Explanation:**

The claims are imprecise or poorly worded, making them difficult to support with the evidence provided. Refinements of the claims are needed to accurately describe what the evidence presents.

**Requested Changes:**

The main issue with the paper contribution claims need to be precise and unambiguously supported by the evidence presented. Please see the weaknesses mentioned for the issues needed to be fixed in order to fix the contribution claims.

---

> ### Author Response · Authors · 2026-07-02
> **Response addressing the comments**
>
> Thank you for pointing out that several contribution claims needed to be made more precise and more directly tied to the evidence. We revised the paper around this point.
>
> ### Delta Rule Definition
>
> You noted that the delta rule was used before being precisely described. We agree. The introduction now defines the shared recurrence early:
>
> `S_t = G_t ⊙ S_{t-1} + κ_t ⊗ u_t`
>
> with `S_0 = 0`, and then defines the delta-rule write as a prediction-error update using the normalized key:
>
> `u_t = v_t - S_{t-1}^T k_hat_t`.
>
> The method section now uses the same notation (Section 3.1), so the variants can be understood in terms of the introduction equations rather than requiring the reader to wait until the technical section.
>
> ### Scope Of The Main Claims
>
> You raised three claim-precision issues, and we revised each.
>
> First, the channel-wise claim is now scoped to the studied language-modeling settings (Abstract; Introduction; Finding 5; Conclusion). We no longer state a universal claim that channel-wise granularity helps only with the delta rule. The revised claim says that, in our LM experiments and in the complete data-independent 2x2 comparison, channel-wise decay hurts without the delta rule but helps with it. We also explicitly discuss the recall exception, where scalar+delta variants outperform channel-wise+delta variants (Section 4.8; Section 5.3).
>
> Second, we removed wording that suggested a formal variance decomposition. The paper now says "strongest observed factor" and defines that phrase using rank separation and matched loss gaps (Finding 4). In the complete data-independent 2x2 subspace, the averaged delta-rule improvement is 0.210 nats on TinyStories and 0.174 nats on WikiText-103, compared with average absolute granularity effects of 0.080 and 0.056. Across matched static-vs-data-dependent comparisons, the average gaps are 0.018 and 0.027. The paper explicitly states that this is not a formal variance decomposition.
>
> Third, we removed the "solves exact recall" wording. The revised paper describes the recall result as a single-seed controlled probe (Section 4.8): DeltaNet and ScalarStaticDelta achieve above 90% mean accuracy under the 15K-step budget, while the channel-wise variants plateau near 27%. We also state that this suggests a task-dependent tradeoff rather than proving a universal limitation.
>
> ### Fixed-Budget Versus Convergence
>
> You noted that fixed-budget comparisons do not establish convergence-optimal rankings. We agree, and this is now explicit throughout the paper. The setup and training-curve discussion state that the experiments compare variants under a matched training budget, not after independently tuning each model to its best validation loss (Sections 4.1 and 4.5). The limitations now include a dedicated paragraph explaining that all curves are still improving at the final checkpoint and that slower-converging variants could close or reverse some gaps under longer or independently tuned training (Section 5.6).
>
> ### Associative Recall Underfitting And Longer-Sequence Analysis
>
> For your concern about the associative recall table (Table 8 in the original submission, now Table 9), we agree that the 15K-step setting alone cannot prove that the lower-accuracy variants would never reach 100% with longer training. We therefore foreground the existing 30K-step budget sweep for KDA and StaticChannelDelta and soften the interpretation (Section 4.8, Table 11). Doubling the budget improves those two variants by only about one percentage point, so the revised paper says the plateau is not resolved by doubling the budget and is consistent with, but does not prove, a structural difficulty.
>
> Separately, you noted that the longer-sequence language-modeling experiment was not tied clearly to the claims. We revised this section as an exploratory stress test rather than primary evidence. The paper now states that the seq_len=1024 result is single-seed, covers only 5 of 8 variants, and is not part of the primary claims (Section 4.11; Conclusion).
>
> ### Presentation Fixes
>
> We also made the requested presentation changes:
>
> - Table 1 now has a self-contained caption defining what "data-dependent" and "data-independent" mean. It explicitly clarifies that "data" refers to the input token representation `x_t`, not the recurrent state `S_{t-1}`.
> - "Uniform" initialization was renamed to "Identical" to avoid confusion with uniformly random initialization.
> - The rank heatmap was regenerated so the colorbar is outside the panels.
> - The recall plot was moved to the appendix as a visual aid while the main text keeps the table with exact values.

---

### Review · Reviewer_AZpC · 2026-06-14

**Summary Of Contributions:**

This paper systematically study how delta rule performs in combination with granularity, conditioning in various settings. The key findings suggest that delta helps with channel-wise granularity and performs as the strongest factor in controlled comparisons. This paper has conducted extensive experiments on two datasets and studied the scaling factors by increasing model parameters.

**Audience:**

No

**Audience Explanation:**

I have some concerns which I explain in the following points:
1. For the model setup, this papers uses GPT2 in three different scales. I think the problem this paper studies is fundamental and should be helpful to a broad models based on the effectiveness. ~125M model could be a base model and larger scales should be experimented. Besides, it would be helpful for us to know how the studies are performing on other transformers as well as other self-attention layers.
2. On the datasets, since this paper is focusing on empirical studies, two text-based datasets seem to be not enough, I wonder authors have any thoughts on extending to multimodal data?
3. To further validate findings, I think it would be helpful to introduce a seed set and calculate mean and std by running through seed set, rather than using only 3 seeds.
4. Overall, to me KDA is a quite strong method across all experiments. The purpose of this paper is not proposing a new method, but I wonder whether authors have insights how to leverage all findings on improving existing methods?

**Broader Impact Concerns:**

No ethical concerns observed.

**Claims And Evidence:**

Yes

**Claims Explanation:**

This paper has provided a clear literature search to explain motivations, background and baseline. Section 3 has explained the flow of how this paper carry out all variants to study in each axis.

In experimental sections, this paper presents extensive studies based on its settings on two major datasets. All variants of setup have been experimented and jointly contribute to various findings.

Authors also illustrate training loss studies, ablation, model scaling as well as deep limitation analysis, which benefit to the community.

**Requested Changes:**

Major changes:
I have some major concerns listed above and hope authors can provide explanations.

Addtional changes:
1. In sections 4.2, 4.3, highlighted explanations would be helpful for each table to guide us understand  key findings.
2. In Figure 4, I did not see "solid orange" line?

MInor changes:
1. duplicate texts in the abstract: "the gap consistently larger than within-group spread (6× at 125M)".
2. In Table 2, add a comment on bold texts.

---

> ### Author Response · Authors · 2026-07-02
> **Response addressing the comments**
>
> Thank you for the comments on scope, scale, presentation, and practical implications. We revised the paper to make the intended contribution clearer: it is a controlled isolation of interaction structure among decay-design choices, not a claim of state-of-the-art performance or final architectural superiority. We believe this controlled isolation is useful to the linear-attention audience even before larger-scale validation, because choices about delta-rule writes, gate granularity, and data-dependent versus data-independent decay recur across current linear-attention architectures.
>
> ### Scale And Architecture Scope
>
> You asked for larger scale and broader architectures. We have clarified that our contribution is a controlled comparison under a shared implementation, not a claim about asymptotic scaling or optimized production systems (Sections 4.1, 4.10, and 5.6). The paper now states that the naive recurrent-loop implementation limits feasible scale, and that validation beyond 125M would require chunkwise kernels. We also avoid claiming that the 125M TinyStories result establishes a cross-dataset scaling law. The conclusions are scoped to the shared GPT-style backbone used in the experiments. We do not test alternative backbone families, hybrid softmax/linear layer placements, or architecture-specific tuning in this revision.
>
> ### Dataset And Downstream Scope
>
> You noted that two text datasets are narrow and asked about broader settings such as multimodal data. We now explicitly state that the empirical scope is language modeling plus one controlled retrieval probe (Section 5.6). We do not make claims about multimodal settings, instruction tuning, downstream transfer, or real-world task utility. Those are listed as future work rather than implied by the current evidence.
>
> ### Number Of Seeds
>
> You noted that three seeds may be underpowered. We now emphasize structural findings over fine-grained within-top-3 rankings. The paper states that the delta-rule vs. no-delta-rule-write grouping is consistent across all seeds, but that the within-top-3 ordering is noisy and 5+ seeds would be needed for formal significance tests (Sections 4.4 and 5.6).
>
> ### Practical Implications
>
> You asked how the findings can improve existing methods. We added an explicit "Implications for Model Design" subsection (Section 5.4). The revised implication is that the delta-rule write should be treated as the default design choice in this space, while scalar-gated and channel-wise-gated delta layers may serve complementary roles: scalar-gated layers for precise binding/retrieval and channel-wise-gated layers for richer multi-timescale representation. We present this as an architectural direction suggested by the controlled comparisons, not as a measured hybrid benchmark.
>
> ### Table And Figure Presentation
>
> We added guiding takeaways and caption notes to the main result tables. Sections 4.2 and 4.3 now include explicit takeaways immediately after the TinyStories and WikiText-103 result tables. Tables 2 and 3 now clarify that bold entries denote new data-independent conditions; Tables 4 and 5 add bold-best conventions; Table 13 highlights the scale-up pattern that delta-rule variants remain separated from models without a delta-rule write. We also fixed the duplicated abstract sentence.
>
> For the training-curve visibility issue, Figure 4 has been regenerated with clearer markers and legend handles so the DeltaNet/orange curve is visible and unambiguous.

---

### Review · Reviewer_4Djq · 2026-06-24

**Summary Of Contributions:**

This paper presents a controlled factorial study of decay mechanisms in linear attention, systematically evaluating the 2×2×2 design space.  he authors implement 8 controlled variants within a shared GPT-style backbone and evaluate them on TinyStories and WikiText-103 at 18/42/125M parameters. The key finding is that the delta rule is the single strongest factor: all 4 delta variants beat all 4 non-delta variants at every tested scale.

**Audience:**

Yes

**Audience Explanation:**

This is a systematic study of factors that affect decay mechanism in linear attention, and this will be interest of a large group of the TMLR's audience.

**Claims And Evidence:**

Yes

**Claims Explanation:**

In general most of the claims are well supported. The paper is comprehensive on identifying the 2x2x2 design space and fill in the gap of the current design of decay mechanism.

**Requested Changes:**

- The cross-dataset validation is a strength at 18M but absent at 125M. Running all 8 variants on WikiText-103 at 125M would significantly strengthen the scale-dependent claims.
- Add at least one established long-range benchmark. Even at small scale, evaluating on a subset of tasks from SCROLLS, PG-19, or a simple needle-in-haystack probe at 2K–4K tokens would bridge the gap between the synthetic recall probe and real-world utility.
- Nit: There is duplicated sentence in the abstract, potentially caused by copy paste error. Please fix that.

---

> ### Author Response · Authors · 2026-07-02
> **Response addressing the comments**
>
> Thank you for the positive assessment and for suggesting additional validation experiments. We agree that the requested experiments would strengthen the paper, but we have not added new experimental results in this revision. Instead, we narrowed the current claims and made the missing validations explicit.
>
> ### 125M Cross-Dataset Validation
>
> You noted that the 125M scale-up is TinyStories-only and asked for WikiText-103 at 125M. We agree that this would be the cleanest cross-dataset scale validation. Since we do not add that experiment in this revision, we revised the wording so the scale-up claim is not cross-dataset. The paper now states that the KDA-vs-StaticChannelDelta reversal at 125M should be interpreted as a TinyStories scale-up result, while cross-dataset validation is supported at the 18M scale (Section 4.10; Section 5.6).
>
> ### Long-Range Or Established Retrieval Benchmarks
>
> You suggested adding an established long-range benchmark or a needle-in-haystack probe at 2K-4K. We agree this would be valuable, and these are the natural next validations as compute and implementation bandwidth allow. Because we do not include a new benchmark in this revision, the paper now frames associative recall as a controlled mechanism probe rather than evidence of downstream or real-world utility (Section 4.8). The limitations explicitly state that truly long-range dependencies and broader tasks remain untested, and name needle-in-haystack, MRCR, and in-context learning as important future evaluations (Section 5.6).
>
> This narrower framing is intended to make the current paper's evidence and claims align: the paper characterizes interaction structure under controlled LM and synthetic-recall settings, but does not claim to establish real-world long-context utility.
>
> ### Duplicate Abstract Sentence
>
> The duplicated abstract sentence has been removed in the revised version.

---

### Author Response · Authors · 2026-06-26
**General response to reviewers**

Thank you to all reviewers for the detailed feedback. We appreciate the comments identifying both the value of the controlled comparison and the places where the manuscript needs clearer scoping.

We are preparing a revision focused on tightening the claims and presentation. In particular, we will clarify that the results are controlled fixed budget comparisons rather than convergence optimal or universal architecture claims; define the delta rule earlier in the paper and make the design space description unambiguous.

We will also clarify the scope of the 125M experiments, the single-seed nature of the recall probe and the limitations around scale, sequence length, datasets, and downstream tasks.

We will respond to each reviewer individually shortly and upload a revised PDF incorporating these changes.